# An Evolutionary Paradigm Favoring Cross Talk between Bacterial Two-Component Signaling Systems

Bharadwaj Vemparala,[a] Arjun Valiya Parambathu,[a]* Deepak Kumar Saini,[b,c] Narendra M. Dixit[a,b]

[a]Department of Chemical Engineering, Indian Institute of Science, Bangalore, Karnataka, India
[b]Centre for Biosystems Science and Engineering, Indian Institute of Science, Bangalore, Karnataka, India
[c]Department of Molecular Reproduction, Development, and Genetics, Indian Institute of Science, Bangalore, Karnataka, India

**ABSTRACT** The prevalent paradigm governing bacterial two-component signaling systems (TCSs) is specificity, wherein the histidine kinase (HK) of a TCS exclusively activates its cognate response regulator (RR). Cross talk, where HKs activate noncognate RRs, is considered evolutionarily disadvantageous because it can compromise adaptive responses by leaking signals. Yet cross talk is observed in several bacteria. Here, to resolve this paradox, we propose an alternative paradigm where cross talk can be advantageous. We envisioned programmed environments, wherein signals appear in predefined sequences. In such environments, cross talk that primes bacteria to upcoming signals may improve adaptive responses and confer evolutionary benefits. To test this hypothesis, we employed mathematical modeling of TCS signaling networks and stochastic evolutionary dynamics simulations. We considered the comprehensive set of bacterial phenotypes, comprising thousands of distinct cross talk patterns competing in varied signaling environments. Our simulations predicted that in programmed environments phenotypes with cross talk facilitating priming would outcompete phenotypes without cross talk. In environments where signals appear randomly, bacteria without cross talk would dominate, explaining the specificity widely seen. Additionally, a testable prediction was that the phenotypes selected in programmed environments would display one-way cross talk, ensuring priming to future signals. Interestingly, the cross talk networks we deduced from available data on TCSs of *Mycobacterium tuberculosis* all displayed one-way cross talk, which was consistent with our predictions. Our study thus identifies potential evolutionary underpinnings of cross talk in bacterial TCSs, suggests a reconciliation of specificity and cross talk, makes testable predictions of the nature of cross talk patterns selected, and has implications for understanding bacterial adaptation and the response to interventions.

**IMPORTANCE** Bacteria use two-component signaling systems (TCSs) to sense and respond to environmental changes. The prevalent paradigm governing TCSs is specificity, where signal flow through TCSs is insulated; leakage to other TCSs is considered evolutionarily disadvantageous. Yet cross talk between TCSs is observed in many bacteria. Here, we present a potential resolution of this paradox. We envision programmed environments, wherein stimuli appear in predefined sequences. Cross talk that primes bacteria to upcoming stimuli could then confer evolutionary benefits. We demonstrate this benefit using mathematical modeling and evolutionary simulations. Interestingly, we found signatures of predicted cross talk patterns in *Mycobacterium tuberculosis*. Furthermore, specificity was selected in environments where stimuli occurred randomly, thus reconciling specificity and cross talk. Implications follow for understanding bacterial evolution and for interventions.

**KEYWORDS** *Mycobacterium tuberculosis*, cross talk, evolutionary dynamics, mathematical modeling, specificity, two-component regulatory systems

Address correspondence to Narendra M. Dixit, narendra@iisc.ac.in.

*Present address: Arjun Valiya Parambathu, Department of Chemical and Biomolecular Engineering, University of Delaware, Newark, Delaware, USA.

The authors declare no conflict of interest.

Bacteria sense and respond to environmental cues predominantly via two-component signaling systems (TCSs) (1). The first component of a TCS is the transmembrane histidine kinase (HK). The HK detects the stimulus, which typically is a chemical ligand, and gets autophosphorylated. The phosphorylated HK (HK-P) binds to and transfers its phosphoryl group to the response regulator (RR), the second component of the TCS. Phosphorylated RR (RR-P) typically dimerizes and triggers changes in downstream gene expression, mounting a response to the stimulus (1, 2). Cognate HK-RR pairs, which belong to a TCS, are generally coexpressed under a single promoter in an operon (3) and are often upregulated as part of the response to the stimulus (1, 2).

Bacteria can have many tens of distinct TCSs, each performing a different function (1). Evolutionary pressure is thought to have rendered TCSs specific: the HK of a TCS rarely phosphorylates the RR of another TCS (4). Cross talk between TCSs, defined as phosphotransfer from the HK of one TCS to the RR of another TCS, is considered disadvantageous because it dissipates the signal, decreasing the concentration of the cognate RR-P and thereby weakening the response (4). Moreover, unwanted responses due to gene expression downstream of noncognate RR-Ps might get triggered. Bacteria typically acquire novel TCSs through gene duplication (5), which would naturally allow cross talk before diversification of the TCSs into distinct pathways (6, 7). Several experimental and modeling studies have argued that despite the extensive homology between TCS proteins, there is strong evolutionary pressure for these paralogs to be specific (5, 8–13). For instance, cross talk between TCSs can be abrogated by as few as two mutations, indicative of the evolutionary pressure favoring specificity (8). Further, during the evolution of new TCSs postgene duplication, bacteria have been predicted to eliminate cross talk before new TCS functionalities can arise (9). The sequence space occupied by the paralogs is thought to be sparse, allowing easy establishment of such specificity (12).

Yet cross talk between bacterial TCSs continues to be observed and, in some bacteria, in significant measure. Approximately 3% of the 850 interactions between TCS proteins in *Escherichia coli*, for instance, were between noncognate HK-RR pairs (14). A substantially larger fraction, ~50% of the 23 interactions, were between noncognate pairs in *Mycobacterium tuberculosis* (15). Given the evolutionary advantages of specificity together with the relative ease of establishing it, the observed cross talk is puzzling. Indeed, in some organisms, such as *Caulobacter crescentus* (16) and *Myxococcus xanthus* (17), no cross talk has been observed among hundreds of interactions. The observed cross talk may thus not be attributable to chance and may instead have evolutionary underpinnings. Unraveling potential evolutionary advantages of cross talk is expected to have important implications for our understanding of bacterial adaptation, survival, and response to interventions (1, 15, 18, 19).

Here, we conceived of an evolutionary paradigm in which cross talk could be beneficial. We hypothesized that in programmed environments, where signals consistently appear in a predefined sequence, cross talk between TCSs that would prime the bacterium to upcoming signals might confer an evolutionary advantage. To test this hypothesis, we constructed a mechanistic mathematical model of generalized multi-TCS signaling networks and performed extensive evolutionary dynamics simulations. We challenged model predictions with available experimental observations and found evidence in support of our hypothesis. Additionally, we arrived at a plausible synthesis of the seemingly conflicting observations of specificity and cross talk in bacterial TCS systems.

## RESULTS

**Cross talk can confer a fitness advantage in programmed environments.** We first considered a hypothetical environment involving $N = 2$ signals, denoted $I_1$ and $I_2$, recognized by two TCSs of a bacterium, $TCS_1$ and $TCS_2$, made up of the proteins $HK_1$ and $RR_1$ and $HK_2$ and $RR_2$, respectively. Depending on the nature of interactions between the TCSs, four phenotypes could exist (Fig. 1a): (i) with no cross talk (phenotype 1), (ii) with cross talk between $HK_1$ and $RR_2$ (phenotype 2), (iii) with cross talk between $HK_2$ and $RR_1$ (phenotype 3), and (iv) with bidirectional cross talk (phenotype 4). We developed a

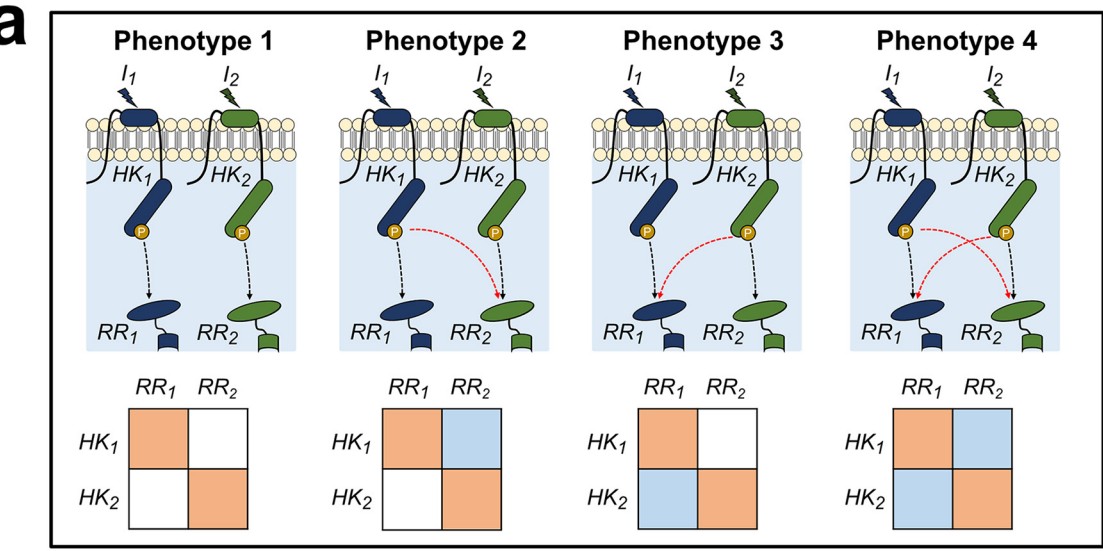

**FIG 1** Mathematical model of TCS signaling predicts advantages of cross talk. (a) All possible phenotypes with $N = 2$ TCSs. Cognate interactions (black arrows) and cross talk (red arrows) are shown. These interactions are also depicted compactly in the interaction matrix for each phenotype (Materials and Methods). Orange squares represent cognate interactions and blue squares represent cross talk. (b) Signal-response behavior and fitness of the phenotypes in a programmed environment. The purple filled rectangles depict

detailed model of signal transduction in a TCS network, allowing for all possible cross talk patterns between the TCSs (Materials and Methods). The model builds on previous models of TCS signaling (9, 15, 20, 21), generalizing them to multi-TCS networks with cross talk. The novelty of our approach lies in recognizing and incorporating the role of the environment. We applied our model to each of the four phenotypes. We first considered the scenario representing a programmed environment. Specifically, we let the signal $I_1$ be followed by $I_2$. For simplicity, we let the signals be identical except for the time of their onset (Fig. 1b). We also assumed the signals to be square pulses arriving in quick succession, mimicking the typical way environments impose stresses (22); we considered alternative signal types below. Using the model, we predicted the concentrations of $RR_1$-P and $RR_2$-P over time (Fig. 1b, top panel) as a proxy for the responses of the bacteria to the two stimuli. Further, we estimated the fitness, $\phi_1$ and $\phi_2$, of the bacteria associated with the responses of the two TCSs and the overall fitness, $\langle\phi\rangle$, combining the two (Fig. 1b, bottom panel). The fitness was determined by the strength of the cognate responses to the individual stimuli (Materials and Methods).

For phenotype 1, where TCSs are insulated, our model predicted that the responses to the two signals were, expectedly, identical except for a shift in time (black curves in Fig. 1b). When $I_1$ arrived, bacterial fitness dropped sharply, indicating a changed environment to which the bacterium was yet to adapt. The bacterium mounted an adaptive response, improving its fitness with time. As $RR_1$-P increased, the fitness, $\phi_1$, recovered. The same phenomenon was observed upon the arrival of $I_2$. The absence of cross talk implied that the responses to $I_1$ and $I_2$ were independent. Although the fitness was nearly fully restored eventually, the time-averaged overall fitness, $\langle\phi\rangle$, was lower than unity, indicative of the vulnerability of the bacterium during adaptation to the changed environment.

For phenotype 2, with $HK_1 \rightarrow RR_2$ cross talk (red curves in Fig. 1b), our model predicted that before the arrival of $I_2$, signal leakage to $TCS_2$ resulted in lower $RR_1$-P and, hence, $\phi_1$ than for phenotype 1. The signal leakage, however, triggered $TCS_2$. The resulting $RR_2$-P upregulated $HK_2$ and $RR_2$. When $I_2$ came up, the bacterium responded faster and better than phenotype 1; $RR_2$-P and $\phi_2$ were higher than for phenotype 1. The overall fitness, $\langle\phi\rangle$, increased beyond that of phenotype 1. Thus, the bacterium was predicted to be more sensitive and responsive to the upcoming stimulus due to cross talk, increasing its fitness. This scenario was illustrative of the possible advantage of cross talk in a programmed environment.

For phenotype 3, with $HK_2 \rightarrow RR_1$ cross talk, in our model predictions, the needless signal dissipation to $RR_1$ following the onset of $I_2$ induced a fitness loss (blue curves in Fig. 1b). Finally, for phenotype 4, with bidirectional cross talk, $RR_1$-P was like phenotype 2 due to dissipation before the arrival of $I_2$. The subtle difference with phenotype 2 arose because of the phosphatase activity of $HK_2$. Cross talk implied that $HK_2$ could exert phosphatase activity on $RR_1$-P, because of which the level of $RR_1$-P was slightly lower and that of $RR_2$-P slightly higher for phenotype 4 than phenotype 2. Thus, immediately upon the arrival of $I_2$, the fitness loss was the least for phenotype 4. However, the advantage of priming was lost due to the $HK_2 \rightarrow RR_1$ cross talk after the arrival of $I_2$, resulting in an overall fitness loss (green curves in Fig. 1b). The predicted time-averaged fitness loss, $1 - \langle\phi\rangle$, of the four phenotypes over the entire signal-response period highlights the advantage of phenotype 2, which has a cross talk pattern that mirrors the signal sequence, over the other phenotypes (Fig. 1b, inset).

**FIG 1** Legend (Continued)

the presence of the input signals, with the darker shade representing $I_1$ and the lighter shade $I_2$. The signal strength is $10^4$ nM for both. The top panel shows the concentrations of activated RRs, and the bottom panel shows the associated fitness of the responses. The phenotypes are color coded, and dark and light curves represent $TCS_1$ and $TCS_2$, respectively. Cross talk strength is $\gamma = 0.26$. The inset shows the reduction in time-averaged fitness of the different phenotypes due to the signals. The fitness is 1 in an unperturbed environment. The fitness of $TCS_1$ when $I_1$ is absent or $TCS_2$ when $I_2$ is absent is thus 1. Note that the fitness curves of all phenotypes in such scenarios overlap. (c) Selection coefficient in a programmed environment. $\sigma$ as a function of $\gamma$ when $I_1$ is followed by $I_2$. (d) Optimal cross talk strength. Dependence of $\sigma$ on $\gamma$ for phenotype 2. The inset shows the fitness of the two TCSs contributing to $\sigma$. (e) Selection coefficients in a random environment. $\sigma$ as a function of $\gamma$ when $I_1$ and $I_2$ follow no order. Fitness is calculated as the mean over all possible signal sequences.

Next, we examined how the fitness advantage would depend on the strength of cross talk using our model. We defined the selection coefficient, $\sigma$, for any phenotype as the difference between the time-averaged fitness of the phenotype and that of phenotype 1, the latter without any cross talk. We quantified the strength of cross talk using $\gamma$, the ratio of the efficiencies of phosphotransfer to noncognate and cognate RRs (Materials and Methods). The larger the value of $\gamma$, the greater was the extent of cross talk. We found from our predictions that for all the values of $\gamma$ studied, phenotype 2 had positive $\sigma$, whereas the other phenotypes had negative $\sigma$ (Fig. 1c), consistent with the results described above. Further, for phenotype 2, $\sigma$ displayed a maximum at intermediate $\gamma$ (Fig. 1d), specifically at $\gamma = 0.26$. Increasing $\gamma$ increased priming and improved the response to $I_2$, increasing fitness. Beyond a point, however, the advantage of priming diminished, but the response to $I_1$ continued to be compromised, lowering the overall fitness (Fig. 1d, inset). Thus, according to our model, limited cross talk offered a fitness advantage to phenotype 2.

**Specificity is advantageous in random environments.** Using the same phenotypes as described above, we applied our model to estimate $\sigma$ in a random environment, where there was no defined sequence of signals (Materials and Methods). Now, phenotype 1 had the highest estimated fitness; $\sigma$ was negative for all the other phenotypes (Fig. 1e). Because the upcoming signal was not prespecified, priming conferred no advantage. The detrimental effects of cross talk then decreased fitness regardless of the cross talk pattern. Thus, $\sigma$ was equal for phenotypes 2 and 3, which had one cross talk interaction each, and lower for phenotype 4, which had two cross talk interactions. Moreover, the greater the value of $\gamma$, the lower was the value of $\sigma$ in the random environment. Thus, in the absence of a consistent sequence of stimuli, our model predicted that evolutionary pressure may select for specificity.

Using sensitivity analysis, we found that the inferences described above were robust to variations in parameter values (see Fig. S1 in the supplemental material). Furthermore, our findings were robust to the fitness construct employed (Text S1; Fig. S2) and the nature of the signals; we tested both square pulses and exponentially decaying signals (Fig. S3). Our model also predicted that with decaying signals, the fitness advantage of cross talk ceased when the interval between the signals was either too small or too large (Fig. S3). When the interval was too small, the second signal appeared before significant priming could happen, whereas when the interval was too large, the priming faded away before the second signal could arrive. These latter predictions were consistent with observations in *E. coli* (23), where priming conferred a significant fitness advantage, manifested as enhanced growth rate, only for a range of time gaps between signals.

**Programmed environments favor one-way cross talk.** For the minimal case of $N = 2$, phenotype 2 alone could anticipate $I_2$ following $I_1$ and thus was predicted to have the highest fitness in our model. For bacteria with more than two TCSs, the fittest phenotype is not obvious, as such anticipation is possible with multiple phenotypes. For instance, the phenotype with the cross talk interactions $HK_1 \rightarrow RR_2$ and $HK_2 \rightarrow RR_3$ as well as the phenotype with $HK_1 \rightarrow RR_2$ and $HK_1 \rightarrow RR_3$ could anticipate the sequence $I_1 \rightarrow I_2 \rightarrow I_3$. The number of phenotypes grows exponentially with $N$. A bacterium with $N$ TCSs will have $N$ cognate and up to $N(N - 1)$ noncognate interactions. Depending on whether each of the latter interactions is realized or not, a total of $2^{N(N-1)}$ phenotypes can exist, each representing a distinct cross talk pattern. For $N = 3$, this would amount to $2^6 = 64$ phenotypes, and for $N = 4$, it would amount to $2^{12} = 4,096$ phenotypes. Identifying the fittest phenotype would thus require a comprehensive assessment of each of these phenotypes. We performed this assessment next.

We considered $N = 3$. We numbered the phenotypes from 1 to 64, starting with the phenotype with no cross talk and ending with the phenotype with all cross talk interactions realized (Fig. 2a). We subjected each phenotype to a programmed environment with the signal sequence $I_1 \rightarrow I_2 \rightarrow I_3$. We also allowed the signals to have different durations, more realistically mimicking natural environments. For each scenario, we applied

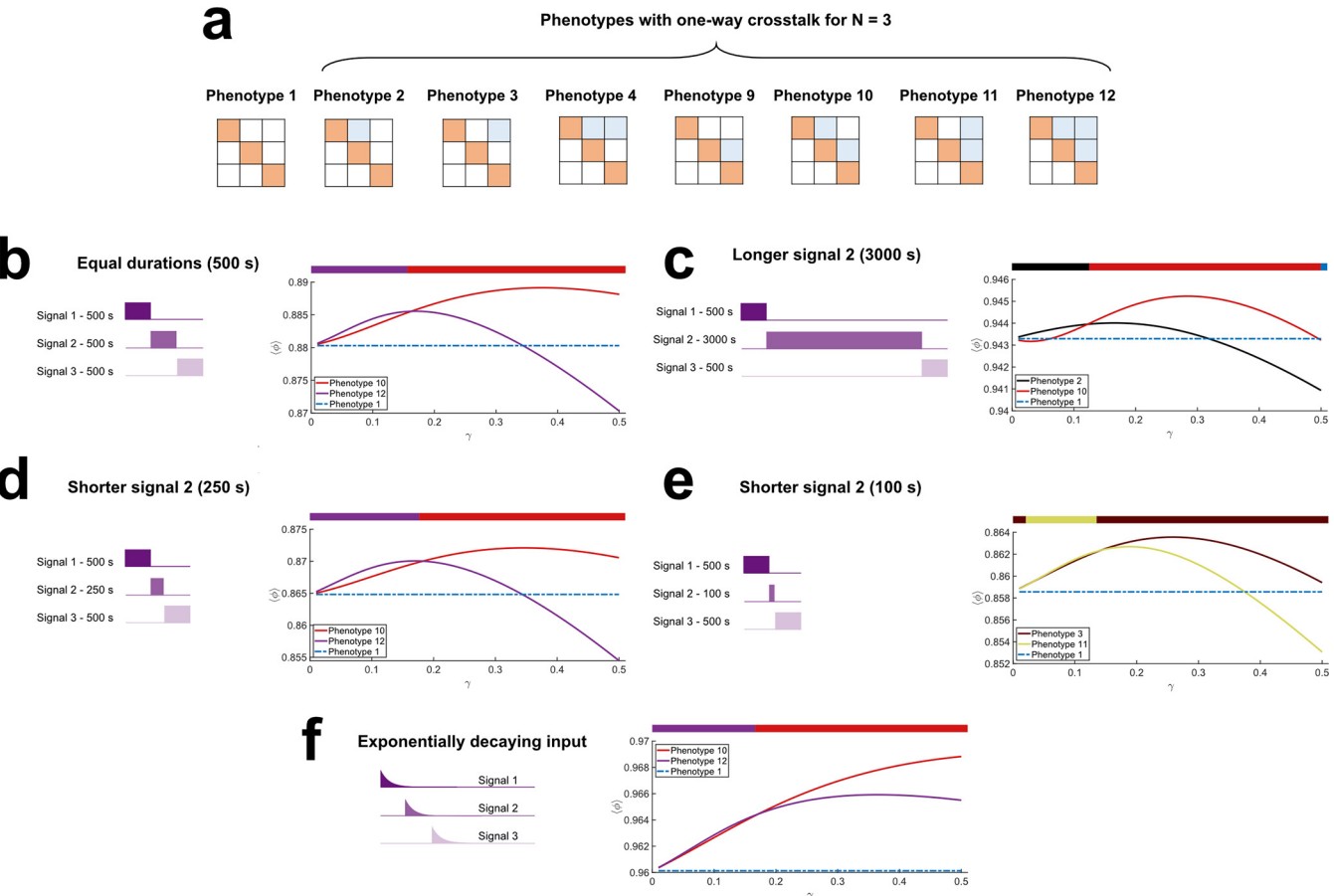

**FIG 2** One-way cross talk patterns yielded the fittest phenotypes. (a) One-way cross talk patterns with $N$ = 3 TCSs. Interaction matrices of phenotype 1, without cross talk, and seven other phenotypes with different one-way cross talk patterns. The signal sequence is $I_1 \rightarrow I_2 \rightarrow I_3$. (b to f) The fitness of the fittest phenotypes and of phenotype 1 as functions of the strength of cross talk, $\gamma$, when (b) signals were of the same duration (500 s) or when $I_2$ lasted (c) 3,000 s, (d) 250 s, and (e) 100 s and (f) when the signals decayed exponentially. The colored bars at the top of each panel graphically depict the range of $\gamma$ over which the respective color-coded phenotype has the highest fitness. Cartoons of the signal patterns are at the left in each panel.

our model to predict signal-response characteristics and estimated the resulting fitness.

When the signals were all of the same duration, our model predicted that the phenotype that was the fittest depended on the strength of cross talk, $\gamma$. When $\gamma$ was small, phenotype 12, which had $HK_1 \rightarrow RR_2$, $HK_2 \rightarrow RR_3$, and $HK_1 \rightarrow RR_3$ interactions, was the fittest (Fig. 2b). Its fitness was only slightly higher than that of phenotype 10, which had $HK_1 \rightarrow RR_2$ and $HK_2 \rightarrow RR_3$ interactions. Note that both these phenotypes anticipated upcoming signals and were fitter than phenotype 1, which had no cross talk. As $\gamma$ increased, phenotype 10 became fitter than phenotype 12 in our predictions. Interestingly, the fitness of the latter decreased beyond a threshold $\gamma$ and eventually dropped below that of phenotype 1. Phenotype 10, however, remained fitter than phenotype 1 throughout. We understood these trends as follows. When $\gamma$ was low, the cost of signal dissipation was small. Thus, the gain from cross talk by $HK_1$ with both $RR_2$ and $RR_3$ and by $HK_2$ with $RR_3$ more than compensated for the fitness loss due to leakage. However, as $\gamma$ increased, the latter cost increased and limiting cross talk became advantageous. Accordingly, our model predicted that cross talk between $HK_1$ and $RR_2$ and between $HK_2$ and $RR_3$, which ensured the requisite anticipation of upcoming signals, was retained, resulting in an overall fitness gain, whereas the redundant cross talk between $HK_1$ and $RR_3$ was eliminated in the fittest phenotype.

We next increased the duration of $I_2$ 6-fold (Fig. 2c). When $\gamma$ was small, phenotype 2, which had the $HK_1 \rightarrow RR_2$ interaction alone, was the fittest in our predictions. As $\gamma$ increased, phenotype 10, which had $HK_1 \rightarrow RR_2$ and $HK_2 \rightarrow RR_3$ interactions, became the

fittest. With weak cross talk, the advantage of priming to $I_3$ through the entire duration of $I_2$ was not enough to compensate for the loss of response to $I_2$. Phenotype 2, which did not have the $HK_2 \rightarrow RR_3$ interaction, was therefore the fittest. On the other hand, when cross talk was stronger, the priming from both $HK_1 \rightarrow RR_2$ and $HK_2 \rightarrow RR_3$ compensated for any signal dissipation, rendering phenotype 10 the fittest in our predictions.

We also considered the effect of shortening the duration of $I_2$ (Fig. 2d and e). When the duration was shortened by 50%, phenotypes 12 and 10 were predicted to be the fittest, depending on $\gamma$, in a manner similar to when the signals were all of the same duration (Fig. 2b and d). The shortening of the duration by 50% thus did not affect the cost-benefit analysis substantially. Shortening the duration 5-fold, however, made a difference, with phenotypes 3 and 11 now the fittest (Fig. 2e). As describe above, when $\gamma$ was small, phenotype 11, with the cross talk interactions $HK_1 \rightarrow RR_3$ and $HK_2 \rightarrow RR_3$, both anticipating the upcoming signal $I_3$, was the fittest in our model. This was because at low values of $\gamma$, priming to $I_3$ while $I_2$ was present did not add significantly to the cost due to signal dissipation, as $I_2$ was present for a short while. However, as $\gamma$ increased, phenotype 3, which had the single cross talk interaction $HK_1 \rightarrow RR_3$, was the fittest. The cost of dissipation, although $I_2$ was short-lived, was no longer affordable. The phenotype that let $I_1$ prime the bacterium to the next major signal, $I_3$, was thus the fittest. Finally, as with the $N = 2$ scenario, the results were similar when exponentially decaying signals were used instead of square pulses (Fig. 2f).

In all these cases, an intriguing feature of the fittest phenotypes is directed, one-way cross talk. If we denote the signal sequence as $I_1 \rightarrow I_2 \rightarrow I_3 \rightarrow \ldots$, then the fittest phenotypes had cross talk of the type $HK_i \rightarrow RR_j$ with $j > i$. In other words, the cross talk that enabled priming to upcoming signals was favored. Reverse signal flow, where $j < i$, resulted in phenotypes that suffered fitness loss. In the interaction matrices, the fittest phenotypes all had nonzero entries in the upper triangular portions and never in the lower triangular portions (Fig. 2a). To test the robustness of this prediction, we adopted two strategies. We performed extensive evolutionary dynamics simulations to examine whether the fitness advantage predicted by the calculations described above would lead to the selection of the corresponding phenotypes with the one-way cross talk patterns. Second, we sought evidence of these predictions in available experimental data.

**Evolutionary simulations predict selection of phenotypes with one-way cross talk patterns mirroring signal sequences.** Using the descriptions mentioned above of the responses of different phenotypes to stimuli, we performed stochastic, discrete generation, Wright-Fisher evolutionary simulations (24) (Fig. 3a; Materials and Methods) to determine which phenotypes would get selected in different environments. We now considered $N = 4$ TCSs, increasing the complexity to a total of 4,096 phenotypes, making it even more difficult to predict the fittest phenotypes intuitively. We performed simulations with two types of initial conditions: (ii) the homogeneous condition, where a single phenotype existed and (ii) the mixed condition, where all the phenotypes were equally represented. With each initial condition, we considered both random and programmed environments. With $N = 4$, we had four types of signals, one for each of the TCSs. We let each bacterium be stimulated four times. In the random environment, each stimulus was chosen randomly from the four possible signals. In the programmed environment, the signals followed a predetermined sequence, where the signals all appeared once and in a fixed order. We computed the fitness of each of the 4,096 species in each of these environments. In each generation, we allowed every bacterium to be selected with a probability proportional to its fitness. The selected bacteria were duplicated to replace lost bacteria and ensure a constant bacterial population. The bacteria were then subjected to mutations. A mutation involved a change in the cross talk network of the bacterium, resulting in an altered phenotype. Specifically, we allowed each of the $N(N - 1) = 12$ potential cross talk interactions within a bacterium to be flipped (from existent to nonexistent and vice versa) with a probability $\mu$, the mutation rate, in each generation. The resulting pool of bacteria formed the substrate for evolution in the next generation. We repeated this process over 10,000 generations, which ensured fixation of the fittest phenotype, and performed 50 realizations, for reliable statistics (Materials and Methods).

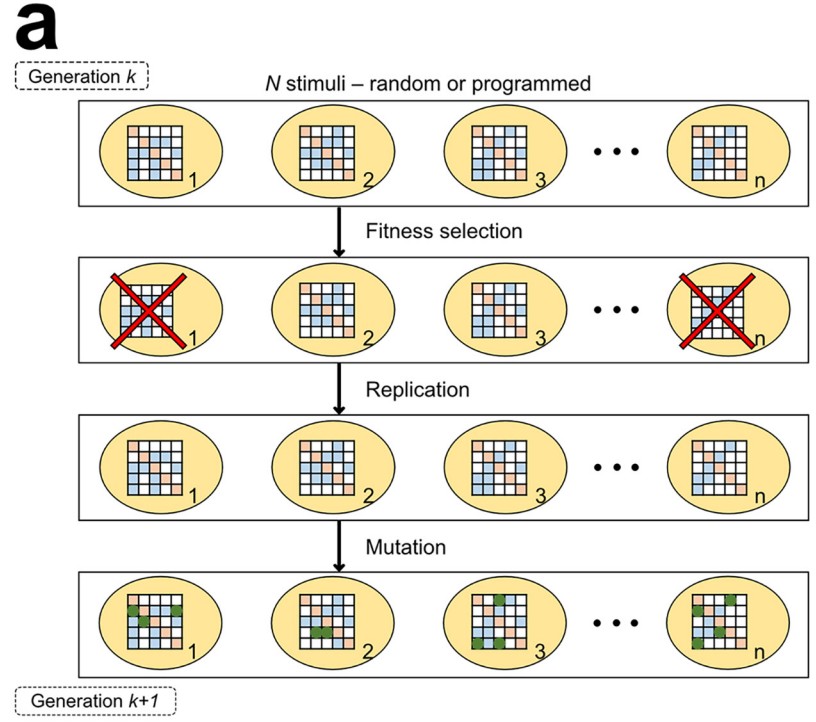

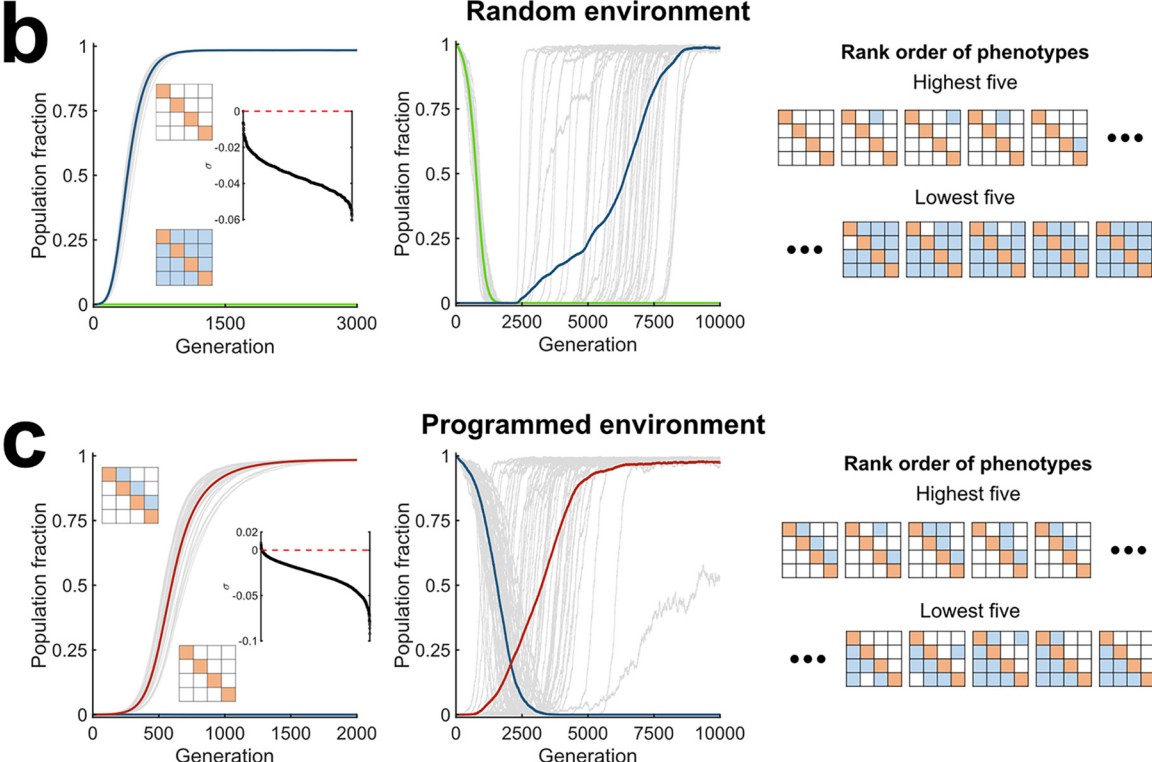

**FIG 3** Stochastic evolutionary dynamics simulations show selection of cross talk in programmed environments and specificity in random environments. (a) Schematic of Wright-Fisher simulations. Simulations proceed in discrete generations and with fixed populations (*n*) comprising bacteria of different phenotypes, indicated by their interaction matrices. In each generation, bacteria are exposed to stimuli. Depending on their response, fitness selection takes place and less fit bacteria are eliminated. Lost bacteria are replaced with copies of selected ones, chosen randomly. The resulting bacteria mutate, illustrated using green boxes in the interaction matrices, resulting in altered phenotypes, which form the substrate for selection in the next generation. (b) Evolution in a random environment. The phenotype without any cross talk (blue) gets fixed whether the initial population is homogeneous (left) or mixed (middle). The phenotype with all cross talk interactions is also shown for comparison (green). The gray lines are trajectories of

In the random environment, our simulations predicted that the phenotype without any cross talk dominated the population (Fig. 3b). For the homogeneous condition, we initiated simulations with the species containing all cross talk interactions. Gradually, phenotypes with fewer cross talk interactions emerged. Eventually, the phenotype with no cross talk emerged and dominated the population. With the mixed condition, the latter species began to dominate the population from the early stages and was soon fixed in the population. These observations agree with the prevalent paradigm of TCS signaling favoring specificity (5, 8, 9, 12). Also, rank-ordering phenotypes by their fitness values (Fig. 3b, inset) revealed that phenotypes with an increasing number of cross talk interactions had decreasing fitness. To illustrate this, we present the cross talk patterns of the top five and bottom five fittest phenotypes (Fig. 3b). The former have zero or one cross talk interaction and the latter have all or one less cross talk interactions.

In the programmed environment, which followed the signal sequence $I_1 \rightarrow I_2 \rightarrow I_3 \rightarrow I_4$, the phenotype with the cross talk pattern mirroring this signal sequence dominated the population (Fig. 3c). For the homogeneous condition, we used the species without cross talk to initiate simulations. Gradually, mutants with cross talk emerged and grew, causing the initial species to decline. Eventually, the phenotype with the cross talk pattern mirroring the signal sequence emerged and dominated the population. For the mixed condition, the latter phenotype grew from the early stages and was rapidly fixed. Arranging the fitness values in descending order (Fig. 3c, inset) displays the benefit of priming for upcoming stimuli. The five fittest phenotypes all had cross talk interactions in the upper triangle of their interaction matrices, indicating one-way cross talk patterns that prime bacteria to upcoming signals (Fig. 3c). The least fit phenotypes had the lower triangle of the interaction matrices populated, indicating cross talk that had signal flows opposite to the sequence of stimuli.

These results were not restricted to $N = 4$ TCSs. With $N = 2$ (Fig. S4) and $N = 3$ TCSs (Fig. S5) as well, the phenotype with no cross talk was selected in random environments, and the phenotype with the cross talk pattern mirroring the sequence of signals was selected in programmed environments.

These simulations thus point to environments where cross talk may be evolutionarily favored. It is possible that such programmed environments may have been the reasons for the selection of the cross talk that is observed in some bacteria. Our model and simulations go beyond offering a plausible explanation of the origins of such cross talk and predict that the cross talk selected is likely to be one-way. We next sought evidence of one-way cross talk patterns in available experimental data.

**Evidence of one-way cross talk in TCSs of *M. tuberculosis*.** In a recent study, cross talk between the TCSs of *M. tuberculosis* has been mapped using *in vitro* assays of phosphotransfer from HKs to all cognate and noncognate RRs (15). Significant cross talk was observed (Fig. 4a), which allowed us to assess signal flows through extended TCS networks. Using the cross talk interactions, we identified all possible signal flows, or cascades, in the TCSs of *M. tuberculosis* as follows. We considered the HK PhoR, for instance, which showed cross talk with the RR DevR (Fig. 4a). DevS, the cognate HK of DevR, further showed cross talk with the RR NarL. NarS, the cognate HK of NarL, did not engage in any cross talk. Thus, when PhoR gets activated, it can transmit a portion of the signal to DevR. Similarly, cross talk of DevS with NarL would transmit some portion of the signal from DevS-DevR to the NarS-NarL system, at which point the signal flow would be terminated. Hence, PhoR-PhoP, DevS-DevR, and NarS-NarL form a cascade

**FIG 3** Legend (Continued)

the two phenotypes in each of 50 realizations. The thick lines are means. Trajectories of all other phenotypes are not shown. The cross talk strength was set to $\gamma = 0.26$. The inset in the left plot is the rank-ordered selection coefficient of all the phenotypes. The interaction matrices of the five most and five least fit phenotypes are shown (right). (c) Evolution in a programmed environment. The one-way cross talk phenotype mirroring the signal sequence $I_1 \rightarrow I_2 \rightarrow I_3 \rightarrow I_4$, which has the highest fitness, dominates the population (red), whether the initial population is homogeneous (left) or mixed (middle). The inset in the left plot is the rank-ordered selection coefficient of all the phenotypes. The interaction matrices of the five most and five least fit phenotypes are depicted (right). Simulations used $N = 4$ TCSs.

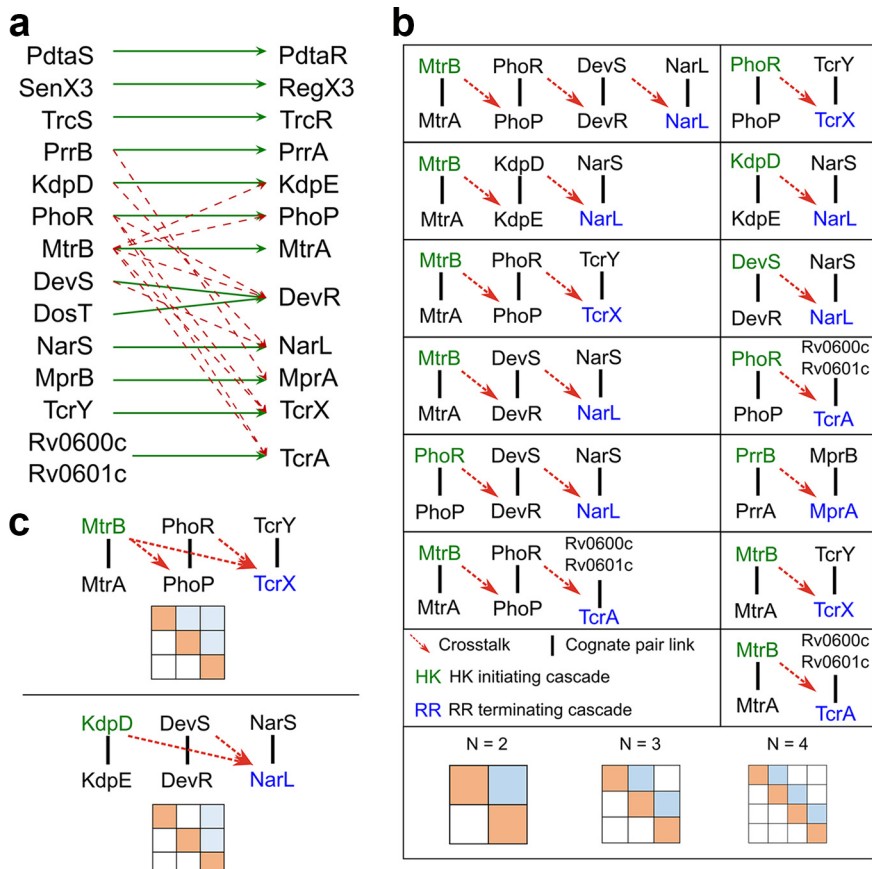

**FIG 4** Cross talk patterns in *M. tuberculosis* TCSs *in vitro* were one-way. (a) Complete cross talk map between TCSs of *M. tuberculosis*. The HKs (left column) and their cognate RRs (right column) are connected by green arrows. The cross talk interactions observed (15) are shown as red dashed arrows. (b) Cross talk cascades. All possible signal flows based on the cross talk interactions in panel a. (c) Superimposed signal cascades. Examples of cross talk patterns resulting from superimposition of cascades from panel b.

of signal flow via cross talk. In this cascade, the signal is not transmitted either to PhoP from DevS or NarS or to DevR from NarS, making the flow one-way.

Following the procedure described above, we started with each of the TCSs of *M. tuberculosis* and traced the resulting cascades. We found 13 such cascades (Fig. 4b). The longest cascade involved 4 TCSs. There were 5 cascades involving 3 TCSs each and 7 cascades involving 2 TCSs each. (Representative interaction matrices for all these cases are presented at the bottom of Fig. 4.) Note that all the cascades had one-way cross talk, with the patterns resembling the fittest phenotypes in our simulations above.

By superimposing the cascades above, we can obtain additional one-way cross talk patterns, reflective of the patterns identified in our simulations. Two such patterns are depicted in Fig. 4c. For instance, the cross talk pattern involving MtrB-MtrA, PhoR-PhoP, and TcrY-TcrX (Fig. 4c, top panel) was equivalent to phenotype 12 in the $N = 3$ case discussed above (Fig. 2b). Similarly, the pattern involving KdpD-KdpE, DevS-DevR, and NarS-NarL (Fig. 4c, bottom panel) was equivalent to phenotype 11 in the $N = 3$ case discussed above (Fig. 2b). Remarkably, we could not find any cross talk pattern that was not one-way. This evidence of exclusive one-way cross talk in the TCSs of *M. tuberculosis* offered support for the predictions of our model and simulations. To assess whether the cross talk could have evolutionarily underpinnings, we sought signatures of evolutionary pressures against diversification post-gene duplication in the sequences of the TCS proteins using bioinformatics analysis (Text S1). Although the analysis

could only be conducted on a subset of the TCSs, the results suggested that this evolutionary pressure may have been less for the TCSs involved in cross talk than for the TCSs that were specific, offering further support to the notion that the observed cross talk may have been evolutionarily favored (Text S1, Fig. S6 and S7, Table S2).

## DISCUSSION

Despite the strong evolutionary arguments favoring specificity in bacterial TCSs (4, 5), cross talk between TCSs has been observed (14, 15). Here, we present an alternative evolutionary paradigm where cross talk would be advantageous. Using modeling of TCS signaling networks and extensive evolutionary dynamics simulations, we predicted that in programmed environments, where stimuli arrive in a predetermined sequence, cross talk that would prime bacteria to upcoming signals would confer an evolutionary benefit. Thus, specific cross talk patterns that mirror the sequences of stimuli could get selected in bacteria living in such environments. Analyzing recent *in vitro* data (15), we found that potential cross talk networks involving the TCSs of *M. tuberculosis* all displayed one-way signal flow, consistent with the notion of priming and selection in programmed environments. This new evolutionary paradigm is not in conflict with the paradigm underlying specificity. Our modeling and simulations predicted that when no predetermined sequence of stimuli existed, specificity was evolutionarily favored. Our study thus offers a conceptual framework that synthesizes specificity and cross talk in bacterial TCS systems. They appear to be two sides of the same coin; they are both outcomes of the same evolutionary forces, but in environments that present signals differently. Programmed environments may be rarer, resulting in the lower prevalence of cross talk.

Independent evidence exists of one-way cross talk aiding bacterial adaptation in programmed environments. In *E. coli*, evolutionary experiments showed how anticipation, facilitated by cross talk, is selected for when the environment displays a specified pattern of carbon source switching (22). Furthermore, the complex structure of environments can become ingrained in *in silico* biochemical networks in order to predict environmental changes preemptively (25). In agreement, this adaptive behavior was evident in *E. coli*, where a match between the covariation of transcriptional responses and the sequence of temperature and oxygen stresses triggering them was observed (25). Evidence also exists of pathogenic bacteria evolving cross talk to adapt to their hosts. For instance, mutations in the TCS BfmS-BfmR of *Pseudomonas aeruginosa* in individuals with cystic fibrosis were recently found to alter, facilitated via cross talk by the noncognate HK GtrS, regulation of downstream gene expression in order to promote biofilm formation and chronic infection (26). Similarly, in *Alphaproteobacteria*, multiple HKs of the HWE/HisKA-2 family can control the phosphorylation of the same response regulators in a coordinated manner and tune downstream gene expression (27).

Based on the signaling cascades we deduced from the *in vitro* TCS cross talk interactions of *M. tuberculosis*, it would be interesting to identify corresponding sequences of stimuli, potentially unveiling information of the environments to which *M. tuberculosis* may have adapted. The ligands/stimuli that many of the TCSs sense, however, remain unknown, precluding such analysis (28). However, specific instances suggesting such adaption could be identified from the cascades. For example, the TCS PrrB-PrrA is reported to be involved in the early replication steps of *M. tuberculosis* inside macrophages (29). The TCS MprB-MprA has been argued to be essential for establishing persistent infection (30), a state of slower or halted replication from which the bacterium can be reactivated to establish active infection (31). Disruption of *mprA* affected processes required for survival during the persistence and subsequent infection stages (30). One could thus argue that cross talk from PrrB-PrrA to MprB-MprA may be favorable because it would prime the bacterium to activate the processes necessary for establishing persistent infection, a key feature of successful tuberculosis infection (32), once entry is gained into a macrophage. Indeed, this one-way cross talk was observed in the

*in vitro* cascades (15). Future experiments may assess the advantage of such cross talk *in vivo*.

Our study has focused on cross talk between HKs and RRs. We recognize that cross talk could also occur at the level of stimuli, where the same stimulus may activate multiple HKs. For instance, the HKs NarX and NarQ of *E. coli* both sense nitrate ions in the environment (33). The extent of the prevalence of such shared stimulation, however, is unknown, as stimuli for many TCSs still remain uncharacterized (28, 34, 35). Nonetheless, although beyond the scope of the present study, our mathematical model can be readily adapted to analyze cross talk arising at the level of stimuli.

Cross talk is not limited to bacterial TCSs. Examples of cross talk exist in human growth factor signaling networks (36), mitogen-activated protein kinase (MAPK) networks of yeast (37), and between TOR and CIP pathways in *Schizosaccharomyces pombe* (38). The evolutionary underpinnings of these cross talk interactions may be more difficult to unravel because of the more involved regulatory structures in these organisms than in the simpler bacterial TCS systems. Yet controlled evolutionary experiments suggest selection of cross-regulation patterns in broad agreement with our predictions. For instance, the yeast *Saccharomyces cerevisiae*, which is commonly used in the fermentation industry, is subjected to heat, ethanolic stress, and oxidative stress, in that order, in the industrial process (22). The related regulatory networks were observed to have the following cross talk interactions: heat $\rightarrow$ ethanolic stress, heat $\rightarrow$ oxidative stress, and ethanolic stress $\rightarrow$ oxidative stress (22). This is similar to the phenotype 12 in the $N = 3$ case in our model (Fig. 2a). Furthermore, when the organism was artificially exposed to these stresses in the reverse order, the cross talk interactions switched their directions (22). These scenarios, together with our proposed paradigm, point to the possible evolutionary advantages of cross talk.

Our findings have implications for the design of signaling systems in synthetic biology. Bacterial TCSs offer promising routes to engineering signaling systems in synthetic biology constructs (39). For instance, they have been used to engineer *E. coli* to sense light (40). Synthetic biology constructs are being designed to sense and integrate multiple stimuli (39). The different TCSs used for such designs are typically assumed to be insulated. However, if the constructs are to be employed in environments that see programmed sequences of the stimuli, then with time, phenotypes that favor cross talk between the TCSs may be selected, potentially affecting the robustness of the constructs. Conversely, where integration of well-defined sequences of stimuli is sought, accounting for the potential selection of phenotypes with cross talk may lead to more robust signaling system designs.

Because of its evolutionary advantages, cross talk may be a potential target of intervention. With pathogenic bacteria, cross talk may sharpen the already sophisticated strategies to evade host immune responses and promote virulence (28, 41). Bacterial HKs offer promising targets of intervention (1, 18). Where cross talk may aid bacterial survival and adaptation, as suggested, for instance, with *M. tuberculosis* (15), targeting HKs engaged in cross talk could prove a more potent strategy than targeting specific HKs. It would not only block the cognate response of the targeted HK, but also compromise the responses of the TCSs that would otherwise have been primed by the targeted HK via cross talk.

## MATERIALS AND METHODS

**Mathematical model of TCS signaling with cross talk.** We developed a mathematical model to describe bacterial signal transduction via TCSs. We considered the scenario in which a bacterium contains $N$ distinct TCSs, which can be engaged in cross talk (Fig. 5a). We built the model by envisioning the set of events associated with the $i$th TCS engaged in cross talk with the $j$th TCS ($i, j \in \{1, 2, \ldots, N\}$), listed below as reactions.

$$\text{HK}_i \underset{k_{b,\text{basal}}}{\overset{k_{f,\text{basal}}}{\rightleftharpoons}} \text{HK}_i^* \tag{1}$$

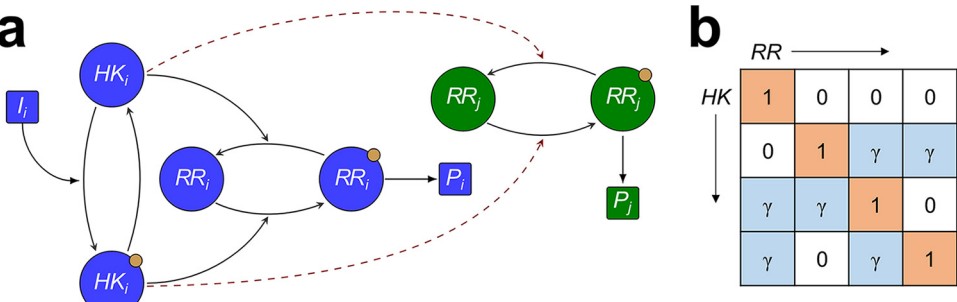

**FIG 5** Schematic of the mathematical model of TCS signaling with cross talk. (a) Architecture of the generalized mathematical model. The input $I_i$ is detected by $HK_i$, which gets phosphorylated ($HK_i$ with a yellow dot) and then transfers the phosphoryl group either to the cognate response regulator, $RR_i$ (blue), or to the noncognate response regulator ($RR_j, j \neq i$ [green]). Activated RRs trigger downstream gene expression via promoter $P_i$. Inactive HKs can act as phosphatases, which dephosphorylate active RRs. (b) Sample interaction matrix for $N = 4$. The diagonal positions represent cognate interactions, and the nondiagonal positions represent noncognate interactions. Zeros in the nondiagonal cells represent the absence of the corresponding cross talk interactions. The ratio of the efficiencies of phosphotransfer to noncognate and cognate interactions is $\gamma$. $2^{N(N-1)}$ such interaction matrices are possible depending on whether each nondiagonal entry is zero or not.

$$I_i + HK_i \underset{k_{b,\text{input}}}{\overset{k_{f,\text{input}}}{\rightleftharpoons}} I_i HK_i \qquad (2)$$

$$I_i + HK_i^* \underset{k_{b,\text{actv,input}}}{\overset{k_{f,\text{actv,input}}}{\rightleftharpoons}} I_i HK_i^* \qquad (3)$$

$$I_i HK_i \underset{k_{b,\text{actv}}}{\overset{k_{f,\text{actv}}}{\rightleftharpoons}} I_i HK_i^* \qquad (4)$$

$$HK_i^* + RR_j \underset{k_{b,ij,\text{phtrf}}}{\overset{k_{f,ij,\text{phtrf}}}{\rightleftharpoons}} HK_i^* RR_j \xrightarrow{k_{\text{phtrf}}} HK_i + RR_j^* \qquad (5)$$

$$I_i HK_i^* + RR_j \underset{k_{b,ij,\text{phtrf}}}{\overset{k_{f,ij,\text{phtrf}}}{\rightleftharpoons}} I_i HK_i^* RR_j \xrightarrow{k_{\text{phtrf}}} I_i HK_i + RR_j^* \qquad (6)$$

$$HK_i + RR_j^* \underset{k_{b,ij,\text{phtse}}}{\overset{k_{f,ij,\text{phtse}}}{\rightleftharpoons}} HK_i RR_j^* \xrightarrow{k_{\text{phtse}}} HK_i + RR_j \qquad (7)$$

$$I_i HK_i + RR_j^* \underset{k_{b,ij,\text{phtse}}}{\overset{k_{f,ij,\text{phtse}}}{\rightleftharpoons}} I_i HK_i RR_j^* \xrightarrow{k_{\text{phtse}}} I_i HK_i + RR_j \qquad (8)$$

$$I_i + HK_i RR_j^* \underset{k_{b,\text{input}}}{\overset{k_{f,\text{input}}}{\rightleftharpoons}} I_i HK_i RR_j^* \xrightarrow{k_{\text{phtse}}} I_i HK_i + RR_j \qquad (9)$$

$$I_i + HK_i^* RR_j \underset{k_{b,\text{actv,input}}}{\overset{k_{f,\text{actv,input}}}{\rightleftharpoons}} I_i HK_i^* RR_j \xrightarrow{k_{\text{phtrf}}} I_i HK_i + RR_j^* \qquad (10)$$

$$2RR_j^* + P_j \underset{k_{p,\text{unbind}}}{\overset{k_{p,\text{bind}}}{\rightleftharpoons}} (RR_j^*)^2 P_j \qquad (11)$$

$$P_j \xrightarrow{k_{btpn}} P_j + m_j \qquad (12)$$

$$(RR_j^*)^2 P_j \xrightarrow{k_{tpn}} (RR_j^*)^2 P_j + m_j \qquad (13)$$

$$m_j \xrightarrow{k_{trn}} m_j + \lambda \cdot HK_j + RR_j \qquad (14)$$

$$I_i \xrightarrow{k_{\text{deg,input}}} \varphi \tag{15}$$

Here, the subscript $i$ refers to the $i$th TCS. We recognize that $HK_i$ can be activated reversibly at some basal level, i.e., in the absence of any input signal, to its active form, $HK_i^*$ (equation 1) (42). The input, $I_i$, can bind reversibly to $HK_i$ or $HK_i^*$ to yield the complexes $I_iHK_i$ or $I_iHK_i^*$, respectively (equations 2 and 3). $I_iHK_i$ can lead to the activated complex $I_iHK_i^*$ at a rate higher than the basal rate above (equation 4). $HK_i^*$ can bind $RR_j$ and activate it via phosphotransfer, yielding $HK_i$ and $RR_j^*$ (equation 5). An analogous reaction occurs with $I_iHK_i^*$ binding to $RR_j$ (equation 6). Note that in these reactions, $j = i$ would imply cognate interactions. $HK_i$ can bind to $RR_j^*$ and exert phosphatase activity (equation 7), consistent with the bifunctional nature of typical HKs, which act as both kinases and phosphatases (1, 9, 43). The latter activity can also be triggered by $I_iHK_i$ (equation 8). The reversible binding of $I_i$ to the intermediate HK-RR complexes is also possible (equations 9 and 10). Thus, we assumed that RR binding to HK does not influence ligand binding to HK. The difference between the efficiencies of activation of cognate and noncognate RRs by a given HK could come from differences in the association rates, dissociation rates, and/or phosphotransfer rates involved. These latter differences are all rarely quantified, although binding affinities and phosphotransfer rates in some select cases have recently been reported (44, 45). Here, for simplicity, we subsumed the differences into the difference in the association rate constants of the HKs with the cognate and noncognate RRs. Specifically, recognizing that the activation rates of noncognate RRs are lower than those of their cognate counterparts, we let the binding rate constants of noncognate partners ($k_{f,ij,\text{phtrf}}$ and $k_{f,ij,\text{phtse}}$) be lower than those of the cognate partners ($k_{f,ii,\text{phtrf}}$ and $k_{f,ii,\text{phtse}}$), with the difference quantified by the attenuation factor $\gamma = \frac{k_{f,ij,\text{phtrf}}}{k_{f,ii,\text{phtrf}}} = \frac{k_{f,ij,\text{phtse}}}{k_{f,ii,\text{phtse}}} < 1$. $RR_j^*$ dimerizes and binds to the corresponding promoter $P_j$ (equation 11). This binding enhances transcription compared to its basal level (equations 12 and 13), i.e., $k_{tpn} > k_{btpn}$, where $k_{btpn}$ and $k_{tpn}$ correspond to basal and activated transcription rate constants, respectively. Transcription produces mRNA, denoted by $m$, which is then translated with the rate constant $k_{trn}$, with the HK and RR translated in the ratio $\lambda$:1 (equation 14). Here, we recognize that the response also typically upregulates the corresponding TCS proteins (2, 46). Input signals degrade with rate constant $k_{\text{deg,input}}$ (equation 15). All the other entities present in the network are assumed to degrade with a rate constant $k_{\text{deg}}$ (not written explicitly for convenience).

Next, we estimated the rate of synthesis of HK and RR proteins by assuming that the DNA binding reactions are fast compared to transcription and translation reactions (15, 20). Let $P_T$ be the total concentration of promoter binding sites present on the bacterial genome, with $f_b$ and $f_f$ the fractions of promoter sites in the bound and free states, respectively. We assumed pseudoequilibrium between DNA binding reactions, yielding

$$k_{p,\text{bind}}(f_f P_T)(RR_j^*)^2 = k_{p,\text{unbind}} f_b P_T \tag{16}$$

If $K_1 = k_{p,\text{unbind}}/k_{p,\text{bind}}$ is the equilibrium dissociation constant for equation 11, we get

$$\frac{f_f}{f_b} = \frac{K_1}{(RR_j^*)^2} \tag{17}$$

Because $f_b + f_b = 1$, it follows that

$$f_f = \frac{1}{1 + \frac{(RR_j^*)^2}{K_1}} \tag{18}$$

and

$$f_b = \frac{1}{1 + \frac{K_1}{(RR_j^*)^2}} \tag{19}$$

We now have the concentration of promoters in the basal and active states. Equations 11 to 13 estimate the rate of upregulation of the corresponding TCS as follows. From equations 12 and 13, the change of mRNA concentration would be

$$\frac{dm_j}{dt} = k_{btpn} f_f P_T + k_{tpn} f_b P_T - k_{\text{deg}} m_j \tag{20}$$

Applying the pseudoequilibrium approximation to mRNA dynamics, i.e., $\frac{dm_j}{dt} \approx 0$, gives

$$m_j = \frac{k_{btpn} P_T}{k_{\text{deg}}} \left( f_f + \frac{k_{tpn}}{k_{btpn}} f_b \right) \tag{21}$$

By substituting equations 18 and 19 into equation 21, we obtain

$$m_j = \frac{k_{btpn} P_T}{k_{\text{deg}}} \frac{\left( 1 + \frac{k_{tpn}}{k_{btpn}} \frac{(RR_j^*)^2}{K_1} \right)}{1 + \frac{(RR_j^*)^2}{K_1}} \tag{22}$$

These mRNA molecules translate at the rate $k_{trn}$ to produce $HK_j$ and $RR_j$ molecules in the ratio $\lambda$:1.

$$\frac{d\mathrm{HK}_j}{dt} = \lambda k_{trn} m_j \tag{23}$$

$$\frac{d\mathrm{RR}_j}{dt} = k_{trn} m_j \tag{24}$$

Substituting $\frac{k_{tpn}}{k_{btpn}} = \alpha$ and $\frac{k_{trn} k_{btpn}}{k_{deg}} = \beta$, we get the synthesis rates of HK and RR by mRNA translation as

$$\frac{d\mathrm{HK}_j}{dt} = \lambda \beta \mathrm{P}_T \cdot \left( \frac{1 + \alpha \cdot \frac{\left(\mathrm{RR}_j^*\right)^2}{K_1}}{1 + \frac{\left(\mathrm{RR}_j^*\right)^2}{K_1}} \right) \tag{25}$$

$$\frac{d\mathrm{RR}_j}{dt} = \beta \mathrm{P}_T \cdot \left( \frac{1 + \alpha \cdot \frac{\left(\mathrm{RR}_j^*\right)^2}{K_1}}{1 + \frac{\left(\mathrm{RR}_j^*\right)^2}{K_1}} \right) \tag{26}$$

The rate equations for equations 1 to 15 can be written following standard mass action terms and by utilizing equations 25 and 26 as follows.

$$\frac{d\mathrm{HK}_i}{dt} = -\left( k_{f,bas} \times \mathrm{HK}_i + k_{f,\mathrm{input}} \times \mathrm{I}_i \times \mathrm{HK}_i + \sum_j k_{f,ij,\mathrm{phtse}} \times \mathrm{HK}_i \times \mathrm{RR}_j^* \right)$$
$$+ \left( k_{b,bas} \times \mathrm{HK}_i^* + k_{b,\mathrm{input}} \times \mathrm{I}_i\mathrm{HK}_i + \sum_j k_{\mathrm{phtrf}} \times \mathrm{HK}_i^*\mathrm{RR}_j + \sum_j k_{b,ij,\mathrm{phtse}} \times \mathrm{HK}_i\mathrm{RR}_j^* \right)$$
$$+ \lambda \beta \mathrm{P}_T \times \left( \frac{1 + \alpha \times \frac{\left(\mathrm{RR}_j^*\right)^2}{K_1}}{1 + \frac{\left(\mathrm{RR}_j^*\right)^2}{K_1}} \right) - k_{\mathrm{deg}} \times \mathrm{HK}_i \tag{27}$$

$$\frac{d\mathrm{HK}_i^*}{dt} = -\left( k_{b,bas} \times \mathrm{HK}_i^* + k_{f,\mathrm{actv,input}} \times \mathrm{I}_i \times \mathrm{HK}_i^* + \sum_j k_{f,ij,\mathrm{phtrf}} \mathrm{HK}_i^*\mathrm{RR}_j \right)$$
$$+ \left( k_{f,bas} \times \mathrm{HK}_i + k_{b,\mathrm{actv,input}} \times \mathrm{I}_i\mathrm{HK}_i^* + \sum_j k_{b,ij,\mathrm{phtrf}} \times \mathrm{HK}_i^*\mathrm{RR}_j \right) - k_{\mathrm{deg}} \times \mathrm{HK}_i^* \tag{28}$$

$$\frac{d\mathrm{I}_i\mathrm{HK}_i}{dt} = -\left( k_{b,\mathrm{input}} \times \mathrm{I}_i\mathrm{HK}_i + k_{f,\mathrm{actv}} \times \mathrm{I}_i\mathrm{HK}_i + \sum_j k_{f,ij,\mathrm{phtse}} \times \mathrm{I}_i\mathrm{HK}_i \times \mathrm{RR}_j^* \right)$$
$$+ \left( k_{f,\mathrm{input}} \times \mathrm{I}_i \times \mathrm{HK}_i + k_{b,\mathrm{actv}} \times \mathrm{I}_i\mathrm{HK}_i^* + \sum_j k_{\mathrm{phtrf}} \times \mathrm{I}_i\mathrm{HK}_i^*\mathrm{RR}_j \right.$$
$$\left. + \sum_j k_{b,ij,\mathrm{phtse}} \times \mathrm{I}_i\mathrm{HK}_i\mathrm{RR}_j^* + \sum_j k_{\mathrm{phtse}} \times \mathrm{I}_i\mathrm{HK}_i\mathrm{RR}_j^* \right) - k_{\mathrm{deg}} \times \mathrm{I}_i\mathrm{HK}_i \tag{29}$$

$$\frac{d\mathrm{I}_i\mathrm{HK}_i^*}{dt} = -\left( k_{b,\mathrm{actv,input}} \times \mathrm{I}_i\mathrm{HK}_i^* + k_{b,\mathrm{actv}} \times \mathrm{I}_i\mathrm{HK}_i^* + \sum_j k_{b,ij,\mathrm{phtrf}} \times \mathrm{I}_i\mathrm{HK}_i^* \times \mathrm{RR}_j^* \right)$$
$$+ \left( k_{f,\mathrm{actv,input}} \times \mathrm{I}_i \times \mathrm{HK}_i^* + k_{f,\mathrm{actv}} \times \mathrm{I}_i\mathrm{HK}_i + \sum_j k_{b,ij,\mathrm{phtrf}} \times \mathrm{I}_i\mathrm{HK}_i^*\mathrm{RR}_j \right) - k_{\mathrm{deg}} \times \mathrm{I}_i\mathrm{HK}_i^* \tag{30}$$

$$\frac{d\mathrm{RR}_j}{dt} = -\left( \sum_i k_{f,ij,\mathrm{phtrf}} \times \mathrm{HK}_i^* \times \mathrm{RR}_j + \sum_i k_{f,ij,\mathrm{phtse}} \times \mathrm{I}_i\mathrm{HK}_i^* \times \mathrm{RR}_j \right)$$
$$+ \left( \sum_i k_{b,ij,\mathrm{phtrf}} \times \mathrm{HK}_i^*\mathrm{RR}_j + \sum_i k_{b,ij,\mathrm{phtrf}} \times \mathrm{I}_i\mathrm{HK}_i^*\mathrm{RR}_j + \sum_i k_{\mathrm{phtse}} \times \mathrm{HK}_i\mathrm{RR}_j^* \right.$$
$$\left. + \sum_i k_{\mathrm{phtse}} \times \mathrm{I}_i\mathrm{HK}_i\mathrm{RR}_j^* \right) + \beta \mathrm{P}_T \times \left( \frac{1 + \alpha \times \frac{\left(\mathrm{RR}_j^*\right)^2}{K_1}}{1 + \frac{\left(\mathrm{RR}_j^*\right)^2}{K_1}} \right) - k_{\mathrm{deg}} \times \mathrm{RR}_j \tag{31}$$

$$\frac{d\mathrm{RR}_j^*}{dt} = -\left( \sum_i k_{f,ij,\mathrm{phtse}} \times \mathrm{RR}_j^* \times (\mathrm{HK}_i + \mathrm{I}_i\mathrm{HK}_i) + k_{p,\mathrm{bind}} \times \left(\mathrm{RR}_j^*\right)^2 \times \mathrm{P}_j \right)$$
$$+ \left( \sum_i k_{\mathrm{phtrf}} \times \left(\mathrm{HK}_i^*\mathrm{RR}_j + \mathrm{I}_i\mathrm{HK}_i^*\mathrm{RR}_j\right) + k_{p,\mathrm{unbind}} \times \left(\mathrm{RR}_j^*\right)^2 \mathrm{P}_j \right.$$
$$\left. + \sum_i k_{b,ij,\mathrm{phtse}} \times \left(\mathrm{HK}_i\mathrm{RR}_j^* + \mathrm{I}_i\mathrm{HK}_i\mathrm{RR}_j^*\right) \right) - k_{\mathrm{deg}} \times \mathrm{RR}_j^* \tag{32}$$

$$\frac{d\text{HK}_i^*\text{RR}_j}{dt} = -\left( \left( k_{f,\text{actv,input}} \times \text{I}_i + k_{b,ij,\text{phtrf}} + k_{\text{phtrf}} \right) \times \text{HK}_i^*\text{RR}_j \right) + \left( k_{b,\text{actv,input}} \times \text{I}_i\text{HK}_i^*\text{RR}_j + k_{f,ij,\text{phtrf}} \times \text{HK}_i^* \times \text{RR}_j \right) - k_{\text{deg}} \times \text{HK}_i^*\text{RR}_j \tag{33}$$

$$\frac{d\text{HK}_i\text{RR}_j^*}{dt} = -\left( \left( k_{f,\text{input}} \times \text{I}_i + k_{\text{phtse}} + k_{b,ij,\text{phtse}} \right) \times \text{HK}_i\text{RR}_j^* \right) + \left( k_{b,\text{input}} \times \text{I}_i\text{HK}_i\text{RR}_j^* + k_{f,ij,\text{phtse}} \times \text{HK}_i \times \text{RR}_j^* \right) - k_{\text{deg}} \times \text{HK}_i\text{RR}_j^* \tag{34}$$

$$\frac{d\text{I}_i\text{HK}_i^*\text{RR}_j}{dt} = -\left( \left( k_{b,\text{actv,input}} + k_{b,ij,\text{phtrf}} + k_{\text{phtrf}} \right) \times \text{I}_i\text{HK}_i^*\text{RR}_j \right) + \left( k_{f,\text{actv,input}} \times \text{I}_i \times \text{HK}_i^*\text{RR}_j + k_{f,ij,\text{phtrf}} \times \text{I}_i\text{HK}_i^* \times \text{RR}_j \right) - k_{\text{deg}} \times \text{I}_i\text{HK}_i^*\text{RR}_j \tag{35}$$

$$\frac{d\text{I}_i\text{HK}_i\text{RR}_j^*}{dt} = -\left( \left( k_{b,\text{input}} + k_{b,ij,\text{phtse}} + k_{\text{phtse}} \right) \times \text{I}_i\text{HK}_i\text{RR}_j^* \right) + \left( k_{f,\text{input}} \times \text{I}_i \times \text{HK}_i\text{RR}_j^* + k_{f,ij,\text{phtrf}} \times \text{I}_i\text{HK}_i \times \text{RR}_j^* \right) - k_{\text{deg}} \times \text{I}_i\text{HK}_i\text{RR}_j^* \tag{36}$$

$$\frac{d\text{P}_j}{dt} = -k_{p,\text{bind}} \times \left( \text{RR}_j^* \right)^2 \times \text{P}_j + k_{p,\text{unbind}} \times \left( \text{RR}_j^* \right)^2 \text{P}_j + k_{\text{deg}} \times \left( \text{RR}_j^* \right)^2 \text{P}_j \tag{37}$$

$$\frac{d\left( \text{RR}_j^* \right)^2 \text{P}_j}{dt} = -k_{p,\text{unbind}} \times \left( \text{RR}_j^* \right)^2 \text{P}_j + k_{p,\text{bind}} \times \left( \text{RR}_j^* \right)^2 \times \text{P}_j + k_{\text{deg}} \times \left( \text{RR}_j^* \right)^2 \text{P}_j \tag{38}$$

$$\frac{d\text{I}_i}{dt} = -k_{\text{deg,input}} \times \text{I}_i \tag{39}$$

The rate constants involved were obtained from the literature (9, 20, 47) (Table S1). The rate equations were integrated in MATLAB using the routine ode15s and with chosen initial conditions (Table S1). In all our simulations, the above-described equations were first solved in the absence of stimuli for a sufficiently long time so that the basal autophosphorylation reactions balanced the degradation reactions and all the proteins reached a steady state. Using the latter as the prestimulus state of the bacterium, the above-described equations were solved in the presence of stimuli. The solution depended on the phenotype, described next.

**Interaction matrix.** For a bacterium with $N$ TCSs, different phenotypes are possible depending on the presence or absence of specific cross talk interactions. An interaction matrix defines the identity of each phenotype (Fig. 5b). The $ij$th element in the matrix represents the strength of the cross-interaction between $\text{HK}_i$ and $\text{RR}_j$ relative to the cognate interaction. The cognate interactions are all assumed to be equally strong and occupy the diagonal entries. The cross-interactions are also assumed to be of the same relative intensity, $\gamma$, whenever they exist. The nondiagonal entities are thus either 0 or $\gamma$. Since there are $N(N-1)$ nondiagonal elements present, with 2 state values possible for each of them, we get $2^{N(N-1)}$ different phenotypes.

**Fitness formulation.** We constructed a fitness variable based on the response of a TCS to a time-dependent input. We defined the fitness corresponding to the $i$th TCS as

$$\phi_i(t) = \exp\left( -\frac{\text{I}_i(t)}{\text{I}_m}(1 - f_b) \right) \tag{40}$$

where $f_b = \frac{1}{1 + \frac{K_1}{(\text{RR}_j^*)^2}}$ follows from equation 19 above. The term $-\text{I}_i(t)/\text{I}_m$ reflects the inverse relationship between the fitness and input intensity. $\text{I}_m$ is taken as the maximum (or peak) input value. Thus, as $\text{I}_i$ increases, it reflects an increasing change in the environment, inducing a more significant fitness loss until the bacterium responds and adapts. The recovery of fitness following the response is determined by the second entity in the fitness variable, $1 - f_b$, where $f_b$ denotes the fraction of promoters bound by RR*. (We recall that $K_1$ is the dissociation constant of $(\text{RR}_j^*)^2\text{P}_j$.) As this fraction increases, the magnitude of the response also rises, leading to greater fitness given the signal. This formulation of fitness makes sure that $\phi_i$ lies between 0 and 1. TCSs are assumed to contribute independently to fitness. Thus, for a bacterium with $N$ TCSs, the total instantaneous fitness is the product of individual fitness values:

$$\phi(t) = \prod_{i=1}^{N} \phi_i(t) \tag{41}$$

In the absence of any signal, $\phi = 1$. Similarly, with a perfect response, i.e., with $f_b = 1$, $\phi$ is again 1. We also considered an alternative fitness formulation and found no qualitative differences in our results (Text S1).

**Stochastic evolutionary simulations.** We performed Wright-Fisher simulations to describe the competition between different phenotypes in random and programmed environments. Such simulations have been used widely to study evolutionary dynamics, including to describe viral diversification and the development of drug resistance (48, 49) and the development of antibody responses following vaccination (50, 51). We considered discrete generations with a fixed population of bacteria. Our simulations had these steps:

1. We initialized the population in one of two ways:

   a. Homogeneous population, comprising a colony of a single, chosen phenotype
   b. Mixed population, comprising equal numbers of all possible phenotypes

2. We computed the fitness of bacteria as follows:

   a. In a programmed environment, we employed the sequence of stimuli $I_1 \rightarrow I_2 \rightarrow \ldots \rightarrow I_N$. The fitness of each phenotype was the time-average of the fitness $\phi(t)$ when all the $N$ signals were elicited once:

$$\langle \phi \rangle = \frac{1}{T} \int_0^T \phi(t)dt \qquad (42)$$

   Here, $T$ was chosen to be the time when the last signal faded away.

   b. In a random environment, the signals were elicited in a random sequence. Thus, $N^N$ signal sequences were possible, allowing the signals to repeat. The fitness of each phenotype was then the mean of its time-averaged fitness estimated separately for each of the $N^N$ possible sequences:

$$\langle \phi \rangle_{\text{sequence}} = \frac{1}{T} \int_0^T \phi(t)dt \qquad (43)$$

$$\langle \phi \rangle = \langle \langle \phi \rangle_{\text{sequence}} \rangle \qquad (44)$$

3. We next estimated control fitness, measuring the fitness in the absence of any response, using:

$$\phi_{\text{control}} = \frac{1}{T} \int_0^T dt \prod_i \exp\left(-\frac{I_i(t)}{I_m}\right) \qquad (45)$$

   This has the same expression as $\phi_i$, but without the $f_b$ term.

4. Fitness selection happens on the bacteria in a generation. For each bacterium, we examined whether the fitness $\langle \phi \rangle$ was larger than $\phi_{\text{control}} + (1 - \phi_{\text{control}}) \times r$, where $r \in [0, 1]$ was a random number from a uniform distribution. The latter choice accounted for any stochastic variations in environmental factors and associated selection forces. If $\langle \phi \rangle$ was larger, the bacterium survived. Otherwise, it was removed. This formalism ensured that bacteria that mounted no responses were not selected and that the rest survived with probabilities proportional to their fitness.

5. From the survivors, we randomly selected, using a uniform random distribution, some bacteria and duplicated them to replace lost bacteria and maintain the population constant. This process assumes that surviving bacteria all have the same ability to multiply.

6. We mutated the resulting bacteria. In our simulations, a mutation toggled a potential cross talk interaction between on and off. For instance, for a bacterium with cross talk between $HK_i$ and $RR_{j'}$ mutation would turn the corresponding $k_{f,ij,\text{phtrf}}$ and $k_{f,ij,\text{phtse}}$ from $\gamma \times 10^{-3}$ nM$^{-1}$ s$^{-1}$ to 0. Every bacterium was checked for the possibility of mutation with probability $\mu$ at each of the $2^{N(N-1)}$ cross talk interactions possible.

7. We repeated the above procedure from step 4.

One generation in our simulation time frame was typically $T = N \times 500$ s, with $N$ signals elicited in each generation. This made sure that all the TCSs could be triggered in principle. We performed simulations over 10,000 generations and over 50 realizations for each parameter setting, which ensured reliable statistics.

**Data availability.** The MATLAB codes used to estimate the fitness values and perform Wright-Fisher simulations and the codon and amino acid sequence files, domain information, alignment files, and raw data for the resulting phylogenetic trees employed for evolution analyses are available at the GitHub repository (https://github.com/narendradixit/TCS_crosstalk_evolution).

## SUPPLEMENTAL MATERIAL

Supplemental material is available online only.
**TEXT S1**, DOCX file, 0.1 MB.
**FIG S1**, TIF file, 0.3 MB.
**FIG S2**, TIF file, 0.5 MB.
**FIG S3**, TIF file, 0.9 MB.
**FIG S4**, TIF file, 1.3 MB.
**FIG S5**, TIF file, 1.8 MB.
**FIG S6**, PDF file, 0.1 MB.

**FIG S7**, TIF file, 0.8 MB.
**TABLE S1**, DOCX file, 0.1 MB.
**TABLE S2**, DOCX file, 0.05 MB.

## ACKNOWLEDGMENTS

We thank Sandhya Visweswariah and Supreet Saini for comments and Gaurav Sankhe for input and discussions.

B.V., A.V.P., and N.M.D. designed the study. B.V. and A.V.P. developed the models and codes and performed calculations. B.V., A.V.P., D.K.S., and N.M.D., analyzed the data. B.V. and N.M.D. wrote the paper. B.V., A.V.P., D.K.S., and N.M.D. edited the paper.

We declare that we do not have any competing interests.

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
