## [Reviewer comments · mSystems]

An evolutionary paradigm favoring crosstalk between bacterial two-component signaling systems

Bharadwaj Vemparala, Arjun Valiya Parambathu, Deepak Saini, and Narendra Dixit

Corresponding Author(s): Narendra Dixit, Indian Institute of Science Bangalore

Review Timeline:

Submission Date:	March 23, 2022
Editorial Decision:	July 3, 2022
Revision Received:	August 23, 2022
Accepted:	September 20, 2022

Editor: Vanni Bucci

Reviewer(s): The reviewers have opted to remain anonymous.

Transaction Report:

DOI: <https://doi.org/10.1128/msystems.00298-22>

July 3, 2022

Prof. Narendra M Dixit
Indian Institute of Science Bangalore
Chemical Engineering
Indian Institute of Science Bangalore
Bangalore 560012
India

Re: mSystems00298-22 (An evolutionary paradigm favoring crosstalk between bacterial two-component signaling systems)

Dear Prof. Narendra M Dixit:

Thank you for submitting your manuscript to mSystems. We have completed our review and I am pleased to inform you that, in principle, we expect to accept it for publication in mSystems. However, acceptance will not be final until you have adequately addressed the reviewer comments.

Preparing Revision Guidelines

Sincerely,

Vanni Bucci

Editor, mSystems

Journals Department
American Society for Microbiology

Reviewer comments:

Reviewer #1 (Comments for the Author):

In this manuscript, Vemparala et al. take a computational approach to addressing the question of how crosstalk in bacterial two-component signaling systems (TCSs) might be evolutionarily favored. Through a combination of deterministic, ODE-based kinetic modeling of TCS networks and stochastic evolutionary simulations, the authors conclude that crosstalk can be beneficial in environments wherein input signals occur in a predictable order. For the crosstalk to be beneficial it must be one-way (not reciprocal), which primes the cells for future exposure to an expected signal. Finally, the authors analyze the known TCS signaling networks of *Mycobacterium tuberculosis* and find that all known cases of crosstalk are indeed one-way.

This work will be of interest to researchers interested in bacterial signal transduction and mathematical modeling. The work is clearly presented and the manuscript well written. The stochastic simulation of evolutionary selection for crosstalk is particularly interesting. The greatest weakness of the work is undoubtedly the lack of experimental evidence for the processes in play, but as a purely computational effort the work is well executed and presented.

I have only minor suggestions for revisions:

1. In Fig. 1b, I do not understand the fitness curves for phenotype 4. Why is the TCS2 fitness at its maximum at $t = 0$? The phenotype 4 curves in general don't seem to follow what is being described in the text.
2. Also in Fig. 1b, For phenotype 4 the fitness loss at the onset of I2 is the least severe out of all 4 phenotypes, but this is not what the text says in line 157.
3. The claims that the one-way crosstalk observed in *Mycobacterium tuberculosis* offers strong support for the model predictions is rather overstated. The observation is compatible with the model, but there are many other ways evolutionary selection for the observed signaling network architecture could have occurred.

Reviewer #2 (Comments for the Author):

In this manuscript, Vemparala et al present an interesting perspective on the emergence of crosstalk in bacterial two-component signalling systems (TCSs), through in silico evolution experiments. They study the response of TCSs in response to various stimuli (random and systematic/'programmed'). Notably, they present evidence for their simulated observations from existing TCSs in *M. tuberculosis*.

Overall, I found the paper to be very well-written, with a solid design of the various computational experiments. They have also carried out sensitivity analyses. All parameters have been justified in Supp. Table 1, and codes have been supplied via GitHub.

I have a few minor comments/questions, which may help strengthen the manuscript further as I list below.

Comments:

1. Line 213, 240: " γ was small" - I am having difficulty understanding the exact (physical) significance of γ , and what low and high values would be. While I understand that γ quantifies crosstalk, and the values possibly range from 0.0 to 1.0, I see that γ is set as 0.26 in many simulations. What is the motivation behind this choice of γ ?
2. Line 754: How were the values of γ estimated?
3. Is it possible to quantify (even if approximately) the γ for *M. tuberculosis* TCS, as an example?
4. Line 285: "large number of generations" - sounds a bit vague. What was the exact number? How was it decided?
5. Line 302: Is there a plausible reason why phenotype without crosstalk dominates the population?
6. The analysis of cross-talk between TCSs in this study is very interesting. Given that there is significant modularity in signalling pathways, and similarity of domains of involved proteins, is it possible that there is crosstalk at the signal level? That is, can the same stimulus have an effect on more than one HK? I presume the model can be readily adapted to study this as well. Perhaps

the authors could remark about this in the Discussion.

7. Line 462: I didn't quite get equations 12-14. m_j is not balanced? I am probably missing something. I also did not seem to find codes corresponding to these equations in `data_set.m`. Also, where would 'm' figure, in Figure 5?

8. Abbreviations in Equations: Almost all 'abbreviations' in the parameter subscripts like $phtrf$ and $phtse$ are self-explanatory, but I was wondering if they could be listed in a Supplementary Table. What's the difference between tpn and trn ?

9. Codes: It is very nice that all the codes have been shared via GitHub. However, it would be useful to have minimal documentation. Perhaps, the `README.md` in each folder can elaborate what is the purpose of each of the scripts, and may be an example "workflow" that can generate one or more of the manuscript figures (or presumably the data for the manuscript figures) would be quite helpful.

10. Line 601: "we randomly selected some": feels vague. Uniformly randomly, I presume. Why not a fitness proportionate selection?

11. Even in Step 4 of the simulations (Line 596), it looks like a threshold for fitness is being applied, rather than fitness proportionate selection. I am not sure what is more reasonable in this setting.

12. Line 611: "large number of generations" and "50 realizations": please specify the number of generations. Also, are 50 realisations sufficient? Was this explored?

13. Based on this study, are there any consequences/learnings for engineering robust signalling systems, in the context of synthetic biology?

14. Line 380: "comprehensive"  "extensive"? (perhaps extensive would be better instead of comprehensive in most places in the paper.)

15. Reference 37 seems incomplete (no page numbers). Italicise the Latin names in the refs. 26-32 (and any others).

In this manuscript, Vemparala et al. take a computational approach to addressing the question of how crosstalk in bacterial two-component signaling systems (TCSs) might be evolutionarily favored. Through a combination of deterministic, ODE-based kinetic modeling of TCS networks and stochastic evolutionary simulations, the authors conclude that crosstalk can be beneficial in environments wherein input signals occur in a predictable order. For the crosstalk to be beneficial it must be one-way (not reciprocal), which primes the cells for future exposure to an expected signal. Finally, the authors analyze the known TCS signaling networks of *Mycobacterium tuberculosis* and find that all known cases of crosstalk are indeed one-way.

This work will be of interest to researchers interested in bacterial signal transduction and mathematical modeling. The work is clearly presented and the manuscript well written. The stochastic simulation of evolutionary selection for crosstalk is particularly interesting. The greatest weakness of the work is undoubtedly the lack of experimental evidence for the processes in play, but as a purely computational effort the work is well executed and presented.

I have only minor suggestions for revisions:

1. In Fig. 1b, I do not understand the fitness curves for phenotype 4. Why is the TCS2 fitness at its maximum at $t = 0$? The phenotype 4 curves in general don't seem to follow what is being described in the text.
2. Also in Fig. 1b, For phenotype 4 the fitness loss at the onset of I2 is the least severe out of all 4 phenotypes, but this is not what the text says in line 157.
3. The claims that the one-way crosstalk observed in *Mycobacterium tuberculosis* offers strong support for the model predictions is rather overstated. The observation is compatible with the model, but there are many other ways evolutionary selection for the observed signaling network architecture could have occurred.

RESPONSE TO REVIEWERS' COMMENTS

on our manuscript (mSystems00298-22) titled

An evolutionary paradigm favoring crosstalk between bacterial two-component signaling systems

We are grateful to both the reviewers for their overall positive reading of our manuscript and their insightful comments. The changes that we have made in response to their comments have enriched and improved our manuscript. Below, we reproduce their comments in blue and our responses in black. Changes made to the manuscript are reproduced in red.

Reviewer #1

In this manuscript, Vemparala et al. take a computational approach to addressing the question of how crosstalk in bacterial two-component signaling systems (TCSs) might be evolutionarily favored. Through a combination of deterministic, ODE-based kinetic modeling of TCS networks and stochastic evolutionary simulations, the authors conclude that crosstalk can be beneficial in environments wherein input signals occur in a predictable order. For the crosstalk to be beneficial it must be one-way (not reciprocal), which primes the cells for future exposure to an expected signal. Finally, the authors analyze the known TCS signaling networks of *Mycobacterium tuberculosis* and find that all known cases of crosstalk are indeed one-way.

This work will be of interest to researchers interested in bacterial signal transduction and mathematical modeling. The work is clearly presented and the manuscript well written. The stochastic simulation of evolutionary selection for crosstalk is particularly interesting. The greatest weakness of the work is undoubtedly the lack of experimental evidence for the processes in play, but as a purely computational effort the work is well executed and presented.

I have only minor suggestions for revisions:

Response: We thank the reviewer for this succinct summary of our manuscript and the overall positive remarks. We address the reviewer's specific concerns below.

1. In Fig. 1b, I do not understand the fitness curves for phenotype 4. Why is the TCS2 fitness at its maximum at $t = 0$? The phenotype 4 curves in general don't seem to follow what is being described in the text.

Response: We apologize for this confusion. The curves for phenotype 4 are not distinct fundamentally from the other phenotypes. In all cases, in the absence of a stimulus, the fitness of the corresponding TCS remains unaffected and stays at unity. Thus, during the time when signal 1 is mounted and signal 2 is yet to be mounted ($t=0$ to $t=500$ s), TCS2 retains its fitness of unity. This is true of all phenotypes in Fig. 1b. The curves for TCS2 of phenotypes 1-3 overlap with those of phenotype 4 during this period (all are unity) and are not seen in the figure. We now clarify this in the caption to Fig. 1b:

“The fitness is 1 in an unperturbed environment. The fitness of TCS₁ when I₁ is absent or TCS₂ when I₂ is absent is thus 1. Note that the fitness curves of all phenotypes in such scenarios overlap.”

2. Also in Fig. 1b, for phenotype 4 the fitness loss at the onset of I₂ is the least severe out of all 4 phenotypes, but this is not what the text says in line 157.

Response: We again apologize for this confusion. The statement on line 157 refers to the overall fitness of phenotype 4. Note that the overall fitness is the time averaged product of the fitness of the individual TCSs. Thus, although the fitness loss of TCS2 right after the onset of I₂ is the least severe for phenotype 4, the advantage is lost subsequently because of the two-way crosstalk in phenotype 4, which results in unnecessary signal dissipation to TCS1 even after I₂ has terminated. We clarify this in the revised manuscript (Lines 137 to 144):

“Finally, for phenotype 4, with bidirectional crosstalk, RR₁-P was like phenotype 2 due to dissipation before the arrival of I₂. The subtle difference with phenotype 2 arose because of the phosphatase activity of HK₂. Crosstalk implied that HK₂ could exert phosphatase activity on RR₁-P because of which the level of RR₁-P was slightly lower and that of RR₂-P slightly higher for phenotype 4 than phenotype 2. Thus, immediately upon the arrival of I₂, the fitness loss was the least for phenotype 4. However, the advantage of priming was lost due to the HK₂→RR₁ crosstalk after the arrival of I₂, resulting in an overall fitness loss (green curves in Fig. 1b).”

3. The claims that the one-way crosstalk observed in *Mycobacterium tuberculosis* offers strong support for the model predictions is rather overstated. The observation is compatible with the model, but there are many other ways evolutionary selection for the observed signaling network architecture could have occurred.

Response: We agree with the reviewer and have toned down the claims of ‘strong’ support throughout. Specifically, we have altered the claims in the abstract and in the results section to indicate that the evidence is consistent with our predictions:

(Lines 38 to 40)

“Interestingly, the crosstalk networks we deduced from available data on TCSs of *Mycobacterium tuberculosis* all displayed one-way crosstalk, ~~offering strong support to which was consistent with~~ our predictions.”

(Lines 325 to 326)

“This evidence of exclusive one-way crosstalk in the TCSs of *M. tuberculosis* offered ~~strong~~ support to the predictions of our model and simulations.”

Reviewer #2

In this manuscript, Vemparala et al present an interesting perspective on the emergence of crosstalk in bacterial two-component signalling systems (TCSs), through in silico evolution experiments. They study the response of TCSs in response to various stimuli (random and systematic/'programmed'). Notably, they present evidence for their simulated observations from existing TCSs in *M. tuberculosis*.

Overall, I found the paper to be very well-written, with a solid design of the various computational experiments. They have also carried out sensitivity analyses. All parameters have been justified in Supp. Table 1, and codes have been supplied via GitHub.

I have a few minor comments/questions, which may help strengthen the manuscript further as I list below.

Response: We are grateful to the reviewer for these kind overall remarks on our paper. We address the reviewer's specific comments and queries below.

1. Line 213, 240: " γ was small" - I am having difficulty understanding the exact (physical) significance of γ , and what low and high values would be. While I understand that γ quantifies crosstalk, and the values possibly range from 0.0 to 1.0, I see that γ is set as 0.26 in many simulations. What is the motivation behind this choice of γ ?

Response: We thank the reviewer for this comment and apologize for the confusion. γ in our model is the ratio of the rates of activation of a non-cognate vs. the cognate response regulator by a phosphorylated histidine kinase. As the reviewer rightly recognizes, it thus quantifies the extent of crosstalk and ranges between 0 and 1. We now further clarify the physical meaning of γ in the revised manuscript (Lines 450-455):

“The difference between the efficiencies of activation of cognate and non-cognate *RRs* by a given *HK* could come from differences in the association rates, dissociation rates, and/or phosphotransfer rates involved. These latter differences are all rarely quantified, although binding affinities and phosphotransfer rates in some select cases have recently been reported (44, 45). Here, for simplicity, we subsumed the differences into the difference in the association rate constants of the *HKs* with the cognate and non-cognate *RRs*.”

On lines 213 and 240, the description refers to Fig. 2, where γ is varied between 0 and 0.5. The phrase “ γ was small” refers to values at the lower end of this range. Such phrasing is used to guide the reader to the different fittest phenotypes that emerged with changing γ .

The choice of $\gamma=0.26$ comes from our calculations for the two TCS scenario ($N=2$) in a programmed environment, presented in Fig. 1d. In these calculations, phenotype 2 had the highest fitness when $\gamma=0.26$. We therefore used this value in many of our subsequent simulations. We now mention this explicitly in the main text while describing Fig. 1d (Line 155):

“Further, for phenotype 2, σ displayed a maximum at intermediate γ (Fig. 1d), specifically at $\gamma=0.26$.”

2. Line 754: How were the values of γ estimated?

Response: In line 754, we point to estimates of the selection coefficient for different values of γ , the latter varied between 0 and 0.2, to illustrate the associated trends in the selection coefficients of the different phenotypes when $N=2$ (presented in Supplementary Fig. 2). The values of γ chosen were to span the range from no crosstalk ($\gamma=0$) to the extent of crosstalk that yielded the maximum fitness gain for phenotype 2 ($\gamma\sim 0.2-0.3$).

3. Is it possible to quantify (even if approximately) the γ for *M. tuberculosis* TCS, as an example?

Response: Such quantification would require data of the kinetics of binding and phosphotransfer of an HK with its cognate as well as a non-cognate RR. Unfortunately, to our knowledge, such data is yet to be generated for *M. tuberculosis* TCSs. Data of the kinetics of the phosphorylation of the same RR by its cognate and a non-cognate HK have been reported (Singh et al., J Mol Biol 431:777-793, 2019). Binding affinities between cognate and non-cognate HK-RR pairs of *M. tuberculosis* TCSs have also been measured (Sankhe et al., bioRxiv doi:10.1101/2021.12.30.474508;2021.12.30.474508). The complete datasets required for estimation of γ , however, are lacking for any HK. The same appears to be the case for other bacteria too. We expect studies in the near future, including from our own

laboratories, to generate these datasets, following which estimates of γ for *M. tuberculosis* TCSs would become available.

4. Line 285: "large number of generations" - sounds a bit vague. What was the exact number? How was it decided?

Response: We agree and have now corrected this. We performed the simulations for 10,000 generations, which was based on the time it took for the fittest phenotype to get fixed in the population. We thus now mention (Lines 264 to 266):

“We repeated this process over 10,000 generations, which ensured fixation of the fittest phenotype, and performed 50 realizations, for reliable statistics (Methods).”

5. Line 302: Is there a plausible reason why phenotype without crosstalk dominates the population?

Response: This is because in the absence of a predefined sequence of stimuli, the advantage of crosstalk, which would prime the bacterium to upcoming stimuli, is lost. Rather, crosstalk would then lead to signal dissipation and thus fitness loss. The phenotype without crosstalk would then dominate the population. To illustrate this, we have presented the phenotypes with the five highest and lowest fitness values in this scenario of randomly occurring stimuli (Fig. 3b right). Indeed, the fittest phenotype is the one with no crosstalk and the least fit is the one with all possible crosstalk interactions realized.

6. The analysis of cross-talk between TCSs in this study is very interesting. Given that there is significant modularity in signalling pathways, and similarity of domains of involved proteins, is it possible that there is crosstalk at the signal level? That is, can the same stimulus have an effect on more than one HK? I presume the model can be readily adapted to study this as well. Perhaps the authors could remark about this in the Discussion.

Response: The reviewer raises a very interesting point. There is at least one known case of the same stimulus activating more than one TCS of a bacterium: The HKs NarX and NarQ of *E. coli* both sense nitrate ions in the environment (Francis and Porter, *Annu Rev Microbiol* 73:199-223). The frequency with which such shared stimulation occurs, however, is unknown, as stimuli for many TCSs remain uncharacterized. Nevertheless, as the reviewer rightly recognizes, our mathematical model can be adapted to analyze crosstalk arising at the level of signals. We have now mentioned this in the Discussion section (Lines 380 to 385):

“Our study has focused on crosstalk between HKs and RRs. We recognize that crosstalk could also occur at the level of stimuli, where the same stimulus may activate multiple HKs. For instance, the HKs NarX and NarQ of *E. coli* both sense nitrate ions in the environment (33). The extent of the prevalence of such shared stimulation, however, is unknown, as stimuli for many TCSs still remain uncharacterized (28, 34, 35). Nonetheless, although beyond the scope of the present study, our mathematical model can be readily adapted to analyze crosstalk arising at the level of stimuli.”

7. Line 462: I didn't quite get equations 12-14. m_j is not balanced? I am probably missing something. I also did not seem to find codes corresponding to these equations in `data_set.m`. Also, where would 'm' figure, in Figure 5?

Response: The equations 12-14 refer to events downstream of RR binding to its promoter. The events involve transcription, both basal (Eq. 12) and upon RR binding to the promoter (Eq. 13), leading to

the production of the corresponding mRNA molecules, m_j . The latter are translated to yield products, which include the TCS proteins HK and RR (Eq. 14). The translation events do not consume the mRNA molecules and thus m_j ‘appear’ unbalanced in the equations. Each mRNA molecule can be translated multiple times before it is degraded by natural cellular degradation processes.

The file `data_set.m` contains the ordinary differential equations 27-39 mentioned in the text, which correspond to the events in the above equations (12-14). In Fig. 5, these transcription and translation events occur downstream of promoter regions P and are not depicted for ease of presentation.

8. Abbreviations in Equations: Almost all 'abbreviations' in the parameter subscripts like `phtrf` and `phtse` are self-explanatory, but I was wondering if they could be listed in a Supplementary Table. What's the difference between `tpn` and `trn`?

Response: We agree and apologize for the lack of clarity with some of the abbreviations. Specifically, `tpn` stands for transcription and `trn` for translation. We have now added a footnote to Supplementary Table 1 explaining all the abbreviations used in the subscripts of the parameter symbols. For the few parameters that do not occur in the table (because they are part of the model development but not the final model equations), we have ensured that their symbols are explained at their first occurrence in the text.

9. Codes: It is very nice that all the codes have been shared via GitHub. However, it would be useful to have minimal documentation. Perhaps, the `README.md` in each folder can elaborate what is the purpose of each of the scripts, and may be an example "workflow" that can generate one or more of the manuscript figures (or presumably the data for the manuscript figures) would be quite helpful.

Response: We thank the reviewer for this suggestion. We have now updated the `README.md` files in our GitHub repository accordingly. It should now be a lot easier to navigate the codes.

10. Line 601: "we randomly selected some": feels vague. Uniformly randomly, I presume. Why not a fitness proportionate selection?

Response: We agree and have now corrected the text as follows (Lines 574 to 575):

“From the survivors, we selected, **using a uniform random distribution**, some and duplicated them to replace lost bacteria and maintain the population constant.”

The reviewer is right in suggesting that fitness proportionate selection could also be used here. In our simulations, fitness proportionate selection is implemented at the level of bacterial survival. Thus, in each generation, bacteria are allowed to survive with a probability proportional to their fitness in the environment (stimuli) they encounter in that generation. The ability to multiply post survival is assumed to be the same for all bacteria. It is based on the latter assumption that we implement the uniform random selection above. It is indeed possible that both survival and the ability to multiply depend on fitness. The effect of fitness selection would thus be further amplified. This would influence our results quantitatively, expediting the fixation of the fittest strains, and not alter our qualitative conclusions of the different phenotypes selected in different environments. Nonetheless, we mention the assumption used in our implementation of fitness selection in the revised manuscript (Lines 575 to 576):

“This process [of uniform random selection] assumes that surviving bacteria all have the same ability to multiply.”

11. Even in Step 4 of the simulations (Line 596), it looks like a threshold for fitness is being applied, rather than fitness proportionate selection. I am not sure what is more reasonable in this setting.

Response: In Step 4, we do implement fitness proportionate selection. The threshold is introduced to make sure that bacteria that do not respond at all to stimuli are not selected. Thus, the threshold is estimated (in Step 3) as the fitness of a bacterium that mounts no adaptive response to stimuli. This fitness is termed $\phi_{control}$. In step 4, bacteria are selected in proportion to their fitness, so long as their fitness is above $\phi_{control}$. We now clarify this in the revised manuscript (Lines 571 to 573):

“This formalism ensured that bacteria that mounted no responses were not selected and that the rest survived with probabilities proportional to their fitness.”

12. Line 611: "large number of generations" and "50 realizations": please specify the number of generations. Also, are 50 realisations sufficient? Was this explored?

Response: We agree and we have now updated this sentence as follows (Lines 584 to 586):

“We performed simulations over 10000 generations and over 50 realizations for each parameter setting, which ensured reliable statistics.”

We did explore the effect of the number of realizations and found that 50 realizations was sufficient. Specifically, we found little difference in our results between 50 and 100 realizations, although subtle deviations were evident with 25 realizations. For illustration, we show comparisons of the evolution of the fraction of the fittest phenotype in two scenarios, averaged over 25, 50, and 100 realizations (Fig. R).

Fig. R. Effect of the number of realizations. Time-evolution of the population fraction of the fittest phenotype with (a) N=2 and (b) N=3 TCSs starting with an equal number of all possible phenotypes in a random environment. The averages were obtained over 25 (blue), 50 (orange) or 100 (yellow) realizations. In (b), the curves overlap.

13. Based on this study, are there any consequences/learnings for engineering robust signalling systems, in the context of synthetic biology?

Response: The reviewer again raises a very interesting point. Yes, there are learnings from our study with implications for designing signaling systems in synthetic biology. We now highlight them in the Discussion (Lines 399 to 408):

“Our findings have implications for the design of signaling systems in synthetic biology. Bacterial TCSs offer promising routes to engineering signaling systems in synthetic biology constructs (39). For instance, they have been used to engineer *E. coli* to sense light (40). Synthetic biology constructs are being designed to sense and integrate multiple stimuli (39). The different TCSs used for such designs are typically assumed to be insulated. However, if the constructs are to be employed in environments that see programmed sequences of the stimuli, then with time phenotypes that favor crosstalk between the TCSs may be selected, potentially affecting the robustness of the constructs. Conversely, where integration of well-defined sequences of stimuli is sought, accounting for the potential selection of phenotypes with crosstalk may lead to more robust signaling system designs.”

14. Line 380: "comprehensive"  "extensive"? (perhaps extensive would be better instead of comprehensive in most places in the paper.)

Response: We agree and have replaced ‘comprehensive’ with ‘extensive’ in the above context and several other places in the revised manuscript.

15. Reference 37 seems incomplete (no page numbers). Italicise the Latin names in the refs. 26-32 (and any others).

Response: We thank the reviewer for noticing these typos. We have now updated and corrected the above references.

September 20, 2022

Prof. Narendra M Dixit
Indian Institute of Science Bangalore
Chemical Engineering
Indian Institute of Science Bangalore
Bangalore 560012
India

Re: mSystems00298-22R1 (An evolutionary paradigm favoring crosstalk between bacterial two-component signaling systems)

Dear Prof. Narendra M Dixit:

Your manuscript has been accepted, and I am forwarding it to the ASM Journals Department for publication. For your reference, ASM Journals' address is given below. Before it can be scheduled for publication, your manuscript will be checked by the mSystems production staff to make sure that all elements meet the technical requirements for publication. They will contact you if anything needs to be revised before copyediting and production can begin. Otherwise, you will be notified when your proofs are ready to be viewed.

Publication Fees:

If you would like to submit a potential Featured Image, please email a file and a short legend to mSystems@asmusa.org. Please note that we can only consider images that (i) the authors created or own and (ii) have not been previously published. By submitting, you agree that the image can be used under the same terms as the published article. File requirements: square dimensions (4" x 4"), 300 dpi resolution, RGB colorspace, TIF file format.

We recognize that the video files can become quite large, and so to avoid quality loss ASM suggests sending the video file via <https://www.wetransfer.com/>. When you have a final version of the video and the still ready to share, please send it to mSystems staff at mSystems@asmusa.org.

Sincerely,

Vanni Bucci
Editor, mSystems

Journals Department
Fig S6: Accept
Fig S2: Accept
Table S1: Accept
Text S1: Accept
Fig S5: Accept
Fig S7: Accept
Fig S1: Accept
Fig S3: Accept
Table S2: Accept
Fig S4: Accept